# DIFFUVST: Narrating Fictional Scenes with Global-History-Guided Denoising Models

**Shengguang Wu, Mei Yuan, Qi Su**[*]
Peking University
{wushengguang,yuanmei}@stu.pku.edu.cn, sukia@pku.edu.cn

## Abstract

Recent advances in image and video creation, especially AI-based image synthesis, have led to the production of numerous visual scenes that exhibit a high level of abstractness and diversity. Consequently, Visual Storytelling (VST), a task that involves generating meaningful and coherent narratives from a collection of images, has become even more challenging and is increasingly desired beyond real-world imagery. While existing VST techniques, which typically use autoregressive decoders, have made significant progress, they suffer from low inference speed and are not well-suited for synthetic scenes. To this end, we propose a novel diffusion-based system DIFFUVST, which models the generation of a series of visual descriptions as a single conditional denoising process. The stochastic and non-autoregressive nature of DIFFUVST at inference time allows it to generate highly diverse narratives more efficiently. In addition, DIFFUVST features a unique design with bi-directional text history guidance and multimodal adapter modules, which effectively improve inter-sentence coherence and image-to-text fidelity. Extensive experiments on the story generation task covering four fictional visual-story datasets demonstrate the superiority of DIFFUVST over traditional autoregressive models in terms of both text quality and inference speed.

## 1 Introduction

Visual Storytelling (VST) is the challenging task of generating a series of meaningful and coherent sentences to narrate a set of images. Compared to image and video captioning (Vinyals et al., 2014; Luo et al., 2020; Wang et al., 2022; Lei et al., 2021), VST extends the requirements of image-to-text generation beyond plain descriptions of images in isolation. A VST system is expected to accurately portray the visual content of each image while also

capturing the semantic links across the given sequence of images as a whole.

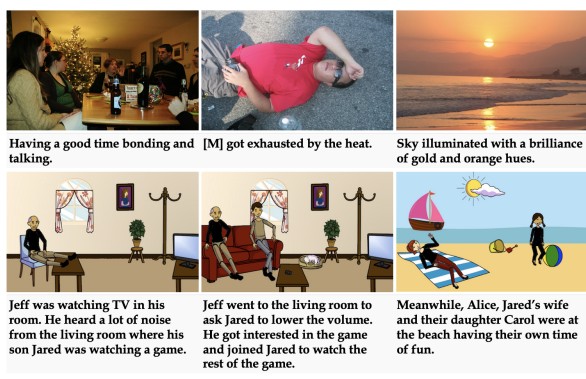

Having a good time bonding and talking.

[M] got exhausted by the heat.

Sky illuminated with a brilliance of gold and orange hues.

Jeff was watching TV in his room. He heard a lot of noise from the living room where his son Jared was watching a game.

Jeff went to the living room to ask Jared to lower the volume. He got interested in the game and joined Jared to watch the rest of the game.

Meanwhile, Alice, Jared's wife and their daughter Carol were at the beach having their own time of fun.

Figure 1: Examples from the VIST (Huang et al., 2016) and AESOP (Ravi et al., 2021) dataset.

So far, prior works on visual storytelling (Wang et al., 2018; Kim et al., 2018; Malakan et al., 2022) typically utilize an autoregressive design that involves an extra sequential model built on top of basic image captioners to capture the relationship among images. More elaborate frameworks with multi-stage generation pipelines have also been proposed that guide storytelling via *e.g.* storyline-planning (Yao et al., 2018), external knowledge engagement (Yang et al., 2019; Hsu et al., 2019, 2020), and concept selection (Chen et al., 2021). Effective as these existing methods are for describing real-world photo streams (*e.g.* the upper row in Figure 1), albeit with their increasingly complex structural designs, their ability to generate stories from fictional scenes (*e.g.* the lower row in Figure 1) remains unverified. In addition, because these models are typically trained to infer in an autoregressive way, they produce captions token by token while conditioning on only previous (unidirectional) history, which prohibits these architectures from refining prefix of sentences based on later generated tokens. As a result, existing autoregressive models are restricted to generating less

---

[*]Corresponding author.

informative narratives at a low inference speed.

To tackle the above issues of VST – especially given the increasing demand for narrating fictional visual scenes – we propose a novel visual storytelling framework DIFFUVST (**Diffu**sion for **V**isual **S**tory**T**elling) based on diffusion models utilizing continuous latent embeddings, which have demonstrated promising capabilities for image synthesis (Rombach et al., 2021; Pan et al., 2022) and language modelling (Hoogeboom et al., 2021; Austin et al., 2021; Li et al., 2022b). Our model generates a set of visual descriptions simultaneously in a non-autoregressive manner by jointly denoising multiple random vectors into meaningful word embeddings conditioned on the image features of all panels (i.e., all image-text pairs in a visual-story sequence). Specifically, DIFFUVST leverages a transformer encoder to learn to restore the word embeddings of the ground-truth story texts from sequences of Gaussian vectors. To enhance the visual-truthfulness of the generated texts, DIFFUVST proposes multimodal feature extractor and adapter modules with pretrained backbone that provide in-domain story-image features as condition and story-text features as classifier-free guidance (Ho and Salimans, 2022). Since these features represent all panels across the entire picture stream, they serve as the "global history" of all the preceding and following frames and texts, which connects the captioning of the current image to other context panels more closely, thus improving the coherence of the whole visual narrative.

We validate the effectiveness of DIFFUVST by performing the visual story generation task on four visual-story datasets with non-real-world imagery: AESOP (Ravi et al., 2021), FlintstonesSV (Maharana and Bansal, 2021), PororoSV (Li et al., 2018), and DiDeMoSV (Maharana et al., 2022) with the latter three converted from Story Visualization (SV) to fictional VST datasets. Quantitative results show that DIFFUVST outperforms strong autoregressive baselines across all four datasets. In addition, DIFFUVST manages to reduce the inference time by a large factor thanks to its non-autoregressive nature.

Our contributions are summarized as follows:

(1) We model the visual narration of a set of images as one single denoising process and propose DIFFUVST, a diffusion-based method for visual storytelling. To our best knowledge, this work is the first to leverage diffusion models and adopt a non-autoregressive approach in visual storytelling.

(2) We propose global-history guidance with adapted multimodal features in DIFFUVST that enhance the coherence and visual-fidelity of the generated stories.

(3) We demonstrate the effectiveness and practical value of our system in face of the surging demand for narrating synthetic scenes by conducting extensive experiments on four fictional visual-story datasets. DIFFUVST outperforms strong autoregressive models in terms of performance while achieving significantly faster inference speeds.

## 2 Related work

### 2.1 Visual storytelling

Visual storytelling (VST) aims to produce a set of expressive and coherent sentences to depict an image stream. Existing work in this area can be broadly divided into two groups of approaches: end-to-end frameworks and multi-stage systems. In the end-to-end pipeline, models are developed to autoregressively generate multi-sentence stories given the image stream in a unified structure (Wang et al., 2018; Kim et al., 2018). Meanwhile, multi-stage approaches that introduce more planning or external knowledge have also shown impressive performance (Yao et al., 2018; Hsu et al., 2020; Chen et al., 2021). Further, some other works are devoted to adopting more elaborate learning paradigms to improve the informativeness and controllability of story generation (Yang et al., 2019; Hu et al., 2019; Jung et al., 2020).

### 2.2 Diffusion models

Diffusion models are a family of probabilistic generative models that first progressively destruct data by injecting noise, and then learn to restore the original features through incremental denoising. Current studies are mostly based on three formulations: Denoising Diffusion Probabilistic Models (DDPMs) (Ho et al., 2020; Nichol and Dhariwal, 2021a), Score-based Generative Models (SGMs) (Song and Ermon, 2019, 2020), and Stochastic Differential Equations (Score SDEs) (Song et al., 2021, 2020), which have shown outstanding performance in image synthesis. Recently, diffusion models have also been employed to address text generation tasks. Either a discrete diffusion method for text data (Austin et al., 2021; He et al., 2022; Chen et al., 2022) is designed, or the word embedding space is leveraged for continuous diffusion (Li et al., 2022b; Gong et al., 2022).

## 3 Background

### 3.1 Diffusion in continuous domain

A diffusion model involves a forward process and a reverse process. The forward process constitutes a Markov chain of latent variables $x_1, ..., x_T$ by incrementally adding noise to sample data:

$$q(x_t|x_{t-1}) = N(x_t; \sqrt{1-\beta_t}x_{t-1}, \beta_t I) \quad (1)$$

where $\beta_t \in (0, 1)$ denotes the scaling of variance of time step $t$, and $x_T$ tends to approximate a standard Gaussian Noise $N(x_t; 0, I)$ if the time step $t = T$ is large enough. In the reverse process, a model parameterised by $p_\theta$ (typically a U-Net (Ronneberger et al., 2015) or a Transformer (Vaswani et al., 2017)) is trained to gradually reconstruct the previous state $x_{t-1}$ from the noised samples at time step $t$, where $\mu_\theta(x_t, t)$ is model's prediction on the mean of $x_{t-1}$ conditioned on $x_t$:

$$p_\theta(x_{t-1}|x_t) = N(x_{t-1}; \mu_\theta(x_t, t), \sum_\theta(x_t, t)) \quad (2)$$

The original training objective is to minimize the negative log-likelihood when generating $x_0$ from the Gaussian noise $x_T$ produced by (1):

$$E_q[-\log(p_\theta(x_0))] \\ = E_q[-\log(\int p_\theta(x_{0:T})\mathrm{d}(x_{1:T}))] \quad (3)$$

Furthermore, a simpler loss function can be obtained following DDPM (Ho et al., 2020):

$$L = \sum_{t=1}^{T} E_{q(x_t|x_0)}\|\mu_\theta(x_t, t) - \hat{\mu}(x_t, x_0)\|^2 \quad (4)$$

where $\hat{\mu}(x_t, x_0)$ represents the mean of posterior $q(x_{t-1}|x_t, x_0)$.

### 3.2 Classifier-free guidance

Classifier-free guidance (Ho and Salimans, 2022) serves to trade off mode coverage and sample fidelity in training conditional diffusion models. As an alternative method to classifier guidance (Dhariwal and Nichol, 2021) which incorporates gradients of image classifiers into the score estimate of a diffusion model, the classifier-free guidance mechanism jointly trains a conditional and an unconditional diffusion model, and combines the resulting conditional and unconditional score estimates.

Specifically, an unconditional model $p_\theta(z)$ parameterized through a score estimator $\epsilon_\theta(z_\lambda)$ is trained together with the conditional diffusion model $p_\theta(z|c)$ parameterized through $\epsilon_\theta(z_\lambda, c)$. The conditioning signal in the unconditional model is discarded by randomly setting the class identifier $c$ to a null token $\varnothing$ with some probability $p_{uncond}$ which is a major hyperparameter of this guidance mechanism. The sampling could then be formulated using the linear combination of the conditional and unconditional score estimates, where $w$ controls the strength or weight of the guidance:

$$\tilde{\epsilon}_\theta(z_\lambda, c) = (1 + w)\epsilon_\theta(z_\lambda, c) - w\epsilon_\theta(z_\lambda) \quad (5)$$

## 4 DIFFUVST

Figure 2 presents the overall architecture and training flow of our model DIFFUVST, which consists of two pretrained multimodal encoders (one for image and one for text) with their separate adapter layers, a feature-fusion module and a transformer-encoder as the denoising learner.

### 4.1 Multimodal encoding with adapters

To obtain the cross-modal features of each image-text panel, we employ a multimodal backbone (in this work: BLIP (Li et al., 2022a)), which serves to provide a solid initial representation of the image stream and its paired text sequences. However, since BLIP is pretrained on large-scale internet data which contains primarily real-world photos, the extracted image- and text- features only represent the most general visual and textual knowledge. Hence, we further propose an adapter module for transferring the BLIP-encoded multimodal features to the specific domain of fictional scenes and texts. Motivated by the adapter design in parameter-efficient transfer learning (Houlsby et al., 2019), we implement the feature-adapters as additional linear layers following the respective last layer of the visual and textual encoders. This mechanism allows the simultaneous exploitation of both the information stored in the pretrained BLIP checkpoint and the freshly learned knowledge from our training data, which leads to the final feature vectors ($F_v$ and $F_t$) that are well adapted for fictional image-text domains.

### 4.2 Feature-fusion for global history condition and guidance

In line with the formulation in Diffusion-LM (Li et al., 2022b), our model DIFFUVST performs visual storytelling as a conditional denoising task in continuous domain. To condition this denoising process (i.e., the generation of story texts) on the

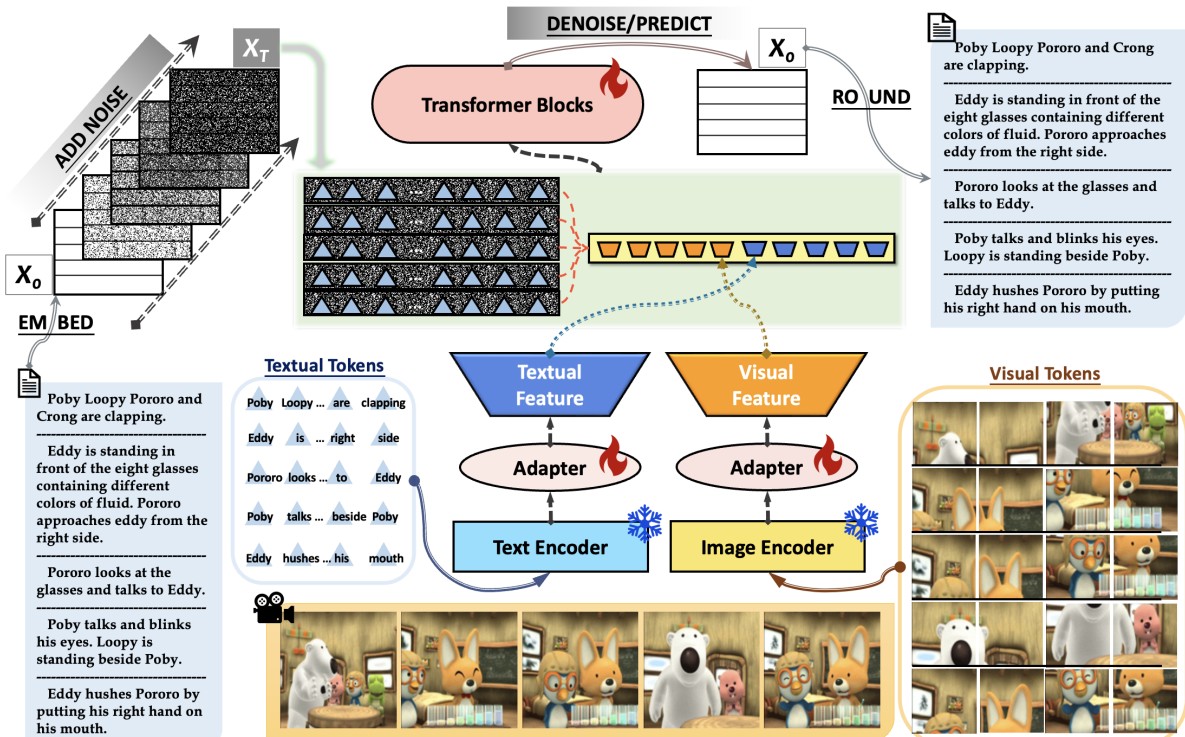

Figure 2: Overview of DIFFUVST: First, two pretrained encoders extract the image- and text features of all panels of the story, which are then transferred to the current dataset domain via trainable adapter modules. Next, these feature segments are fused into the corrupted word embeddings $x_T$ to provide global history condition (image) and classifier-free guidance (text) across the entire picture stream. Finally, a transformer learns to restore the word embeddings of the story ($x_0$) from the fused vector input, which can be grounded back to the actual text.

corresponding visual scenes, we utilize the adapted visual features of all images $\left[F_v^1, F_v^2, ..., F_v^N\right]$ in each story sample as the condition signal, where $N$ is the number of images in a single story sample. For simplicity, we further denote the features of all image panels $\left[F_v^1, F_v^2, ..., F_v^N\right]$ and text panels $\left[F_t^1, F_t^2, ..., F_t^N\right]$ in a visual story sample as $F_v^*$ and $F_t^*$. Specifically, we concatenate them with the corrupted input embeddings $x_T$ to get $[x_T; F_v^*]$. Since these feature segments contain the visual information of all panels in the storyline, they provide global visual history across the entire image stream and thus serve as a more comprehensive condition for coherent storytelling.

Furthermore, we adapt the classifier-free guidance mechanism (Section 3.2) to include a certain degree of ground-truth textual information during training. We do this by fusing the textual features of each panel of the ground-truth story into the above mixed vectors as $[x_T; F_v^*; F_t^*]$, which serves as the final input to our denoising model. However, we mask out the textual feature segment $F_t^*$ with a probability of $p_{unguide}$ (i.e., such textual guidance is provided with a chance of $1 - p_{unguide}$). In

accordance with $p_{uncond}$ in the original classifier-free guidance proposal (Ho and Salimans, 2022), $p_{unguide}$ regulates the percentage of training without extra textual condition that will be unavailable during inference.

Formally, based on the classifier-free guidance mechanism in (5), our DIFFUVST model samples with the feature guidance of global textual history, as expressed in:

$$\tilde{\epsilon}_\theta([x_T; F_v^*]; F_t^*)$$
$$= (1 + w)\epsilon_\theta([x_T; F_v^*]; F_t^*) - w\epsilon_\theta([x_T; F_v^*])$$
$$(6)$$

where $w$ controls the guidance strength of the global textual history. We further discuss the effects of the guidance strength $w$ and unguided probability $p_{unguide}$ in detail in Section 5.4.

Just like the global visual features, these additional signals from the target texts also represent the global history of the whole set of panels in a story sample – but from a textual angle. Hence, such guidance contributes to the bidirectional attention for text generation and improves the overall fluency and coherence of the generated story.

## 4.3 Transformer-encoder for denoising

Our model adopts a BERT-like transformer-encoder architecture to perform the reconstruction of the ground-truth word embeddings $x_0$ (of all the consecutive texts) from a random noise $x_T$. This noisy input is enriched with the global visual- and textual-history features ($[x_T; F_v^*; F_t^*]$) as described in the above fusion module (Section 4.2). We add an additional segment embedding layer to distinguish the adapted BLIP-features from the corrupted sentence embedding $x_T$.

Our model is trained to directly predict the ground-truth word embeddings $x_0$ from any diffusion timestep $t$ sampled from a large $T$ in the same spirit with Improved DDPM (Nichol and Dhariwal, 2021b) and Diffusion-LM (Li et al., 2022b). Instead of using a large $T$, we sample a subset of size $T'$ from the total $T$ timesteps to serve as the actual set of diffusion time steps: $S = s_0, ..., s_{T'}|s_t < s_{t-1} \in (0, T]$. This greatly accelerates the reverse process, which then requires significantly fewer denoising steps, as evidenced in Section 5.3. We also add a $x_1$ restoring loss term $\|\mu_\theta(x_1, 1) - \hat{\mu}(x_1, x_0)\|$ as an absolute error loss (L1-loss) to the L1-version of the basic continuous diffusion loss $L$ as defined in (4) in order to regulate the performance of model on restoring $x_0$ from a slightly noised $x_1$ and to improve the model's stability of prediction. This gives $L'$ as the natural combination of the $x_1$ restoring loss and the L1-version of $L$. Furthermore, we add a rounding term $L_R$ parameterized by $E_q[-\log p(W|\hat{x})] = E_q[-\log \prod_{i=1}^{l} p(W_i|\hat{x}_i)]$ for rounding the predicted $x_0$ embeddings back to discrete tokens, where $W$ is the ground truth sentence and $l$ the generated sequence length. $\hat{x}$ represents the predicted word embeddings.

Our final loss function is the sum of the restoring $x_0$-embedding loss $L'$ and the rounding loss $L_R$, with a rounding term coefficient $\lambda$ controlling the relative importance of $L'$ and $L_R$:

$$
\begin{aligned}
L_{final} &= L' + \lambda L_R \\
&= \sum_{t \in S}^{T'} E_{q(x_t|x_0)}[\|\mu_\theta(x_t, t) - \hat{\mu}(x_t, x_0)\| \\
&\quad + \|\mu_\theta(x_1, 1) - \hat{\mu}(x_1, x_0)\| \\
&\quad - \lambda \log p_\theta(w|\hat{x})]
\end{aligned} \tag{7}
$$

In this work, we set the rounding term coefficient ($\lambda$) as a dynamic $\lambda$ strategy where $\lambda$ is updated after every gradient descent step to ensure that the relative weight of $L'$ and $L_R$ is equal (i.e., $\frac{L'}{L_R} = 1$). This encourages both loss items in (7) to decrease at a relatively same rate.

The predicted embedding $x_0$ is then projected through a language model head to produce all panels of text sequences in parallel that finally form a narrative together.

## 5 Experiments

### 5.1 Experimental setup

**Datasets:** We use four visual-story datasets featuring synthetic style and content as our testbed: AESOP (Ravi et al., 2021), PororoSV (Li et al., 2018), FlintstonesSV (Maharana and Bansal, 2021) and DiDeMoSV (Maharana et al., 2022). It is worth noting that while AESOP was originally designed as an image-to-text dataset, the other three datasets were initially created for story visualization (SV) tasks. However, in our work, we employ them in a reverse manner as visual storytelling datasets due to their fictional art style, which aligns well with the objective of narrating non-real-world imagery. For more detailed statistics regarding these datasets, please refer to Appendix A.1.

**Evaluation metrics:** We evaluate the quality of generated stories using quantitative NLG metrics: BLEU, ROUGE-L, METEOR and CIDEr. We also compare the inference speed of different methods with the average generation time of each test sample using the same hardware and environment.

**Baselines:** Considering the accessibility and reproducibility of existing methods on our four fictional visual storytelling datasets, we compare the performance of our diffusion-based model to three widely recognized open-source autoregressive baselines: GLAC-Net (Kim et al., 2018), TAPM (Yu et al., 2021), BLIP-EncDec (Li et al., 2022a), which are retrained on our four fictional visual storytelling datasets respectively:

- **GLAC-Net** (Kim et al., 2018) employs a LSTM-encoder-decoder structure with a two-level attention ("global-local" attention) and an extra sequential mechanism to cascade the hidden states of the previous sentence to the next sentence serially.

- **TAPM** (Yu et al., 2021) uses a pretrained visual encoder and language generator aligned with Adaptation Loss and finetuned with Sequential Coherence Loss. For our task, we utilize a pretrained ViT (Dosovitskiy et al.,

| Dataset | Method | B@1 | B@4 | M | R | C | Inference Time (sec.) | Denoising Step |
|---|---|---|---|---|---|---|---|---|
| *F.* | GLACNet | 44.55 | 11.98 | 37.48 | 29.23 | 8.93 | 1.2042 | - |
| | BLIP-EncDec | 48.75 | 16.32 | 42.25 | 35.3 | 15.94 | 2.0486 | - |
| | TAPM | 49.05 | 15.08 | 42.12 | 33.03 | 16.54 | 1.8923 | - |
| | DIFFUVST $_{unguided}$ | 48.75 | 16.32 | 42.25 | 35.30 | 15.94 | **0.0873** | 8 |
| | DIFFUVST $_{guided}$ | **49.76** | **17.01** | **42.98** | **36.22** | **17.63** | 0.0981 | 9 |
| *P.* | GLACNet | 23.66 | 2.1 | 21.19 | 17.67 | 2.8 | 0.7844 | - |
| | BLIP-EncDec | 31.38 | 2.96 | 25.13 | 21.64 | 5.63 | 1.8702 | - |
| | TAPM | 28.43 | **3.78** | 24.43 | 21.87 | 5.55 | 1.2289 | - |
| | DIFFUVST $_{unguided}$ | 30.12 | 2.94 | 24.81 | 21.64 | 5.32 | **0.1084** | 10 |
| | DIFFUVST $_{guided}$ | **32.09** | 2.81 | **25.35** | **22.03** | **6.18** | 0.1196 | 11 |
| *D.* | GLACNet | 20.28 | 1.78 | 16.36 | 15.15 | 7.15 | 0.3030 | - |
| | BLIP-EncDec | 20.02 | 2.14 | 16.01 | 17.74 | 14.7 | 0.7474 | - |
| | TAPM | 20.79 | 2.12 | 16.35 | 17.68 | **15.26** | 0.4457 | - |
| | DIFFUVST $_{unguided}$ | 20.90 | 1.88 | 16.41 | 18.77 | 12.51 | **0.0059** | 2 |
| | DIFFUVST $_{guided}$ | **22.08** | **2.74** | **17.33** | **19.75** | 15.16 | **0.0161** | 5 |
| *A.* | GLACNet | 20.37 | 1.02 | 19.46 | 13.39 | 2.14 | 0.9285 | - |
| | BLIP-EncDec | 19.16 | **1.81** | 21.88 | 16.71 | 2.89 | 1.3141 | - |
| | TAPM | 21.95 | 1.56 | 21.55 | 16.3 | 2.97 | 1.1629 | - |
| | DIFFUVST $_{unguided}$ | 9.20 | 0.66 | 15.05 | 13.9 | 0.83 | **0.1982** | 21 |
| | DIFFUVST $_{guided}$ | **23.13** | 1.41 | **21.14** | **19.07** | **3.4** | 0.1988 | 29 |

Table 1: NLG evaluation results as well as the average inference time of our proposed model DIFFUVST and the autoregressive baselines (GLACNet (Kim et al., 2018), BLIP-EncDec (Li et al., 2022a), TAPM (Yu et al., 2021)) on four fictional visual-story datasets in story generation ("Writer" mode) tasks. The suffix *unguided* refers to guidance strength $w = 0$, where our model receives no global textual history guidance at all during training. On the other hand, the *guided* versions are our DIFFUVST models trained with the best configuration of $w$ and $p_{unguide}$ depending on the datasets. *F.*, *P.*, *D.*, and *A.* respectively stand for the abbreviations of the FlintstonesSV, PororoSV, DiDeMoSV, and AESOP datasets. As shown here, our DIFFUVST outperforms various autoregressive baselines with few marginal exceptions, while requiring a significantly lower inference time.

2020) as the visual encoder and GPT2 (Radford et al., 2019) for the language generator, ensuring a fair comparison, as our DIFFUVST also employs a ViT encoder (BLIP-Image-Model) for image features (Section 4.1).

- **BLIP-EncDec** (Li et al., 2022a) leverages the pretrained BLIP captioning model (Li et al., 2022a), fine-tuned on the image-text pairs from our datasets. During inference, we incorporate generated captions from previous images in the textual history of the story sequence, introducing a sequential dependency for captioning the current image. Given that BLIP-EncDec builds upon a well-established captioning model, it serves as a strong baseline, with the mentioned modification specifically for visual storytelling.

We follow the hyperparameter settings specified in each of the baselines and utilize their official codes for training on our four fictional datasets.

**DIFFUVST implementation details:** The multimodal image- and text- feature extractors of DIF-FUVST are realized as branches of the pretrained

BLIP (Li et al., 2022a) and we keep their parameters frozen. Our denoising transformer is initialized with the pretrained *DISTILBERT-BASE-UNCASED* (Sanh et al., 2019). To keep the ground-truth story-text embeddings ($x_0$) stable for training the denoising model, we keep the pretrained word embedding layers and language model head frozen throughout the training process. More detailed hyperparameters for of DIFFUVST's training and inference can be found in Appendix A.2.

### 5.2 Text generation quality and efficiency

The best metric performances of DIFFUVST in comparison to the baselines are reported in Table 1. With regards to the key guidance-related hyperparameters $w$ and $p_{unguide}$, the best results are achieved with $w = 0.3/0.7/1.0/0.3$ and $p_{unguide} = 0.5/0.7/0.7/0.5$ for FlintstonesSV/PororoSV/DiDoMoSV/AESOP datasets respectively. As evident from Table 1, DIFFUVST consistently outperforms these baselines across nearly all metrics and datasets. While there are marginal exceptions, such as the slightly lower B@4 score on the PororoSV and ASEOP datasets and the CIDEr score on DiDeMoSV compared to

TAPM, these do not diminish DIFFUVST's overall superior performance. Notably, given TAPM (Yu et al., 2021) has outperformed most state-of-the-art visual storytellers in their results on VIST, including AREL (Wang et al., 2018), INet (Jung et al., 2020), HSRL (Huang et al., 2019), etc. (as indicated in the TAPM paper), we have good reason to believe that our model can also surpass these models, further demonstrating the effectiveness of DIFFUVST. Moreover, the guided versions of DIFFUVST consistently outperform the fully-unguided (guidance strength $w = 0$) counterparts, which proves our adaptation of classifier-free guidance (Ho and Salimans, 2022) for textual history guidance as highly effective.

In Table 1, we also compared the efficiency of story generation by measuring the average inference time for a single sample on a NVIDIA-Tesla-V100-32GB GPU. Since visual storytelling contains multiple image-text panels, our diffusion-based system that produces all tokens of all text panels simultaneously is able to generate story at a much faster (about 10 times or more) rate than the autoregressive baselines. This acceleration greatly enhances the efficiency of visual storytelling.

In short, DIFFUVST not only achieves top-tier performance (especially with global history guidance), but also excels in efficiency, making it an overall superior choice for visual storytelling.

### 5.3 Inference speed and denoising steps:

The inference time of DIFFUVST is, however, positively correlated with the number of denoising steps, the increment of which is likely to result in a more refined story output. Figure 4 illustrates the change of evaluation results of DIFFUVST with respect to the number of denoising steps. Apparently, our model does not require many steps to reach a stable performance and surpass the autoregressive (AR)-baseline (GLACNet as a demonstrative example), which is a benefit of our model formulation that directly predicts $x_0$ rather than intermediate steps $x_{T-1}$ from the random noise $x_T$. This property further speeds up the story generation, since our model can already produce decent results in only a few denoising steps.

Also noteworthy from Figure 4 is that more denoising steps could result in worse performance. Although a larger number of denoising steps typically enhances generation quality, as is the usual case in image synthesis, we believe an excessive amount of denoising iterations may lead to less satisfactory text generation, especially from the perspective of NLG metrics such as CIDEr. We attribute this phenomenon to the over-filtering of noise. Our denoising transformer is designed to predict ground-truth text embeddings $x_0$ from any noisy inputs $x_t$. Thus, even a single denoising step can effectively filter out a substantial amount of noise. When this filtering process is applied for too many iterations than necessary, however, it continuously tries to reach a "less noisy" text representation and may inadvertently remove valuable linguistic details and result in more uniform and less varied text. And metrics like CIDEr are known to lay greater weight on less common $n$-grams, which explains why the degrading of performance is the most noticeable on the CIDEr metric.

| | BLEU@4 / CIDEr | | |
|---|---|---|---|
| $w$ | $p = 0.3$ | $p = 0.5$ | $p = 0.7$ |
| 0.0 | 1.88 / 12.51 | 1.88 / 12.51 | 1.88 / 12.51 |
| 0.3 | 2.09 / 10.91 | 2.45 / 13.02 | 2.56 / 14.62 |
| 0.5 | 2.23 / 12.03 | 2.51 / 13.97 | 2.68 / 14.08 |
| 0.7 | 2.39 / 10.73 | 2.54 / 11.48 | 2.56 / 12.14 |
| 1.0 | 2.13 / 9.43 | 2.72 / 13.99 | **2.74 / 15.16** |

Table 2: BLEU@4 and CIDEr metric results of DIFFU-VST trained with different configurations of guidance strengths $w$ and unguided probability $p_{unguide}$ (shortended as $p$ in this table) on DiDeMoSV. $w = 0$ refers to fully unguided models and corresponds to DIFFUVST $_{unguide}$ in Table 1. Clearly, using both a high $p_{unguide}$ and high $w$ yields the best result with our DIFFUVST.

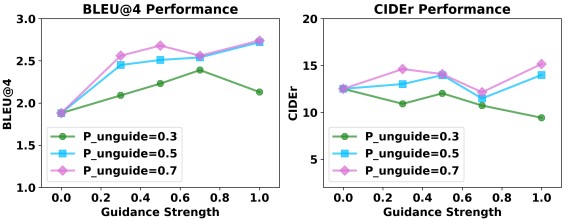

Figure 3: BLEU@4 and CIDEr metric performance of DIFFUVST with varying configurations of guidance strengths $w$ and unguided probability $p_{unguide}$ on DiDe-MoSV. Each curve represents a value of unguided probability $p_{unguide}$. This figure accompanies Table 2. Our diffusion-based model benefits the most from a low guidance rate and strong guidance weight.

### 5.4 Textual feature guidance

We have also experimentally verified the effectiveness of our adaptation of classifier-free guidance (Ho and Salimans, 2022) to including textual

features as additional cues for training our DIF-FUVST model, but with specific probabilities and strengths. The key hyperparameters here are the unguided probability $p_{unguide}$ and the guidance strength $w$. The former regulates the probability of training the model solely on visual representations, without any textual cues (i.e., not guided by global textual history). The latter controls the weighting of the guided model, as shown in (6).

In Table 2 and Figure 3, we take DiDeMoSV as an example dataset and show the varying test performances of our DIFFUVST model when trained with different configurations of guidance strength $w \in \{0.0, 0.3, 0.5, 0.7, 1.0\}$ and unguided probability $p_{unguide} \in \{0.3, 0.5, 0.7\}$. If $w = 0.0$, the model is essentially trained without textual feature guidance at all, which corresponds to the unguided version of DIFFUVST in Table 1.

As Table 2 and Figure 3 show, DIFFUVST achieves the best BLEU@4 and CIDEr results on DiDeMoSV with a strong guidance weight $w = 1.0$ and low guidance rate (i.e., high unguided probability $p_{unguide} = 0.7$). Depending on $p_{unguide}$, the increase of textual guidance strength during training does not necessarily lead to a better model performance, whereas a higher unguided probability consistently produces superior results - regardless of the strength $w$. With a larger $p_{unguide}$, the model is trained to concentrate more on the visual features to learn to tell a story instead of relying too much on the ground-truth textual cues, which will be unavailable at inference time. However, if DIFFUVST receives no textual guidance at all ($w = 0.0$), it may not satisfactorily produce a fluent narration, because the model has to ignore the global textual context altogether, which is also unwanted for storytelling. This is especially demonstrated by the BLEU@4 metric where a fully-unguided model obtains the lowest score.

Based on these observations, we conclude that a relatively small portion of ground-truth textual guidance is ideal for training our DIFFUVST model. Too much textual signals would not fully develop the model's cross-modal ability to caption images into texts and too little conditioning supervision from the story context tends to harm the overall text generation quality. Interestingly, our conclusion also resonates with the findings reported by Ho and Salimans (2022), which emphasize that the sample quality would benefit the most from a small amount of unconditional classifier-free guidance.

| DIFFUVST | B@1 | B@4 | M | R | C |
|---|---|---|---|---|---|
| Dataset = *FlintstonesSV* | | | | | |
| Adapter(+) | **49.76** | **17.01** | **42.98** | **36.22** | **17.63** |
| Adapter(-) | 47.6 | 16.17 | 41.65 | 35.41 | 13.09 |
| Dataset = *PororoSV* | | | | | |
| Adapter(+) | **32.09** | **2.81** | **25.35** | **22.03** | **6.18** |
| Adapter(-) | 31.01 | 2.58 | 25.06 | 21.75 | 5.15 |
| Dataset = *DiDeMoSV* | | | | | |
| Adapter(+) | **22.08** | **2.74** | **17.33** | **19.75** | **15.16** |
| Adapter(-) | 21.1 | 1.72 | 16.93 | 17.62 | 9.19 |
| Dataset = *AESOP* | | | | | |
| Adapter(+) | **23.13** | **1.41** | **21.14** | **19.07** | **3.4** |
| Adapter(-) | 20.52 | 1.28 | 19.75 | 16.84 | 2.42 |

Table 3: Evaluation results across metrics of our DIFFUVST with/without (+/-) Adapter layers under the same hyperparameter settings, including $\lambda$, guidance strength, and guidance probability. The "Adapter(+)" versions are identical to "DIFFUVST $_{guided}$" reported in Table 1. Evidently, the adapters play an important role in boosting our DIFFUVST's performance.

## 5.5 Multimodal feature adapter

In this subsection, we discuss the effects of the adapter modules that transfer the image-text features extracted from the pretrained encoders. The motivations for including the adapter modules are two-fold: Firstly, the adapters serve to transfer the pretrained image-text features to the domain of fictional scenes; Secondly, since these multimodal features are utilized to provide guidance for the optimization of the denoising language model, we need the additional trainable parameters introduced in the adapter modules to better align the encoded features with the capabilities of the denoising LM, thereby enhancing the overall framework integrity of DIFFUVST.

To empirically validate the significance of the adapter modules within our DIFFUVST system, we present here the ablation studies where we compare the performance of our model with and without the adapter layers. The results on four datasets are shown in Table 3. We keep all other hyperparameter settings, including $\lambda$, guidance strength, and guidance probability identical. The "Adapter(+)" versions are thus identical to "DIFFUVST $_{guided}$" reported in Table 1.

As vividly demonstrated in Table 3, the presence of adapter modules significantly impacts model performance across all evaluation metrics and datasets, where DIFFUVST built without adapters consis-

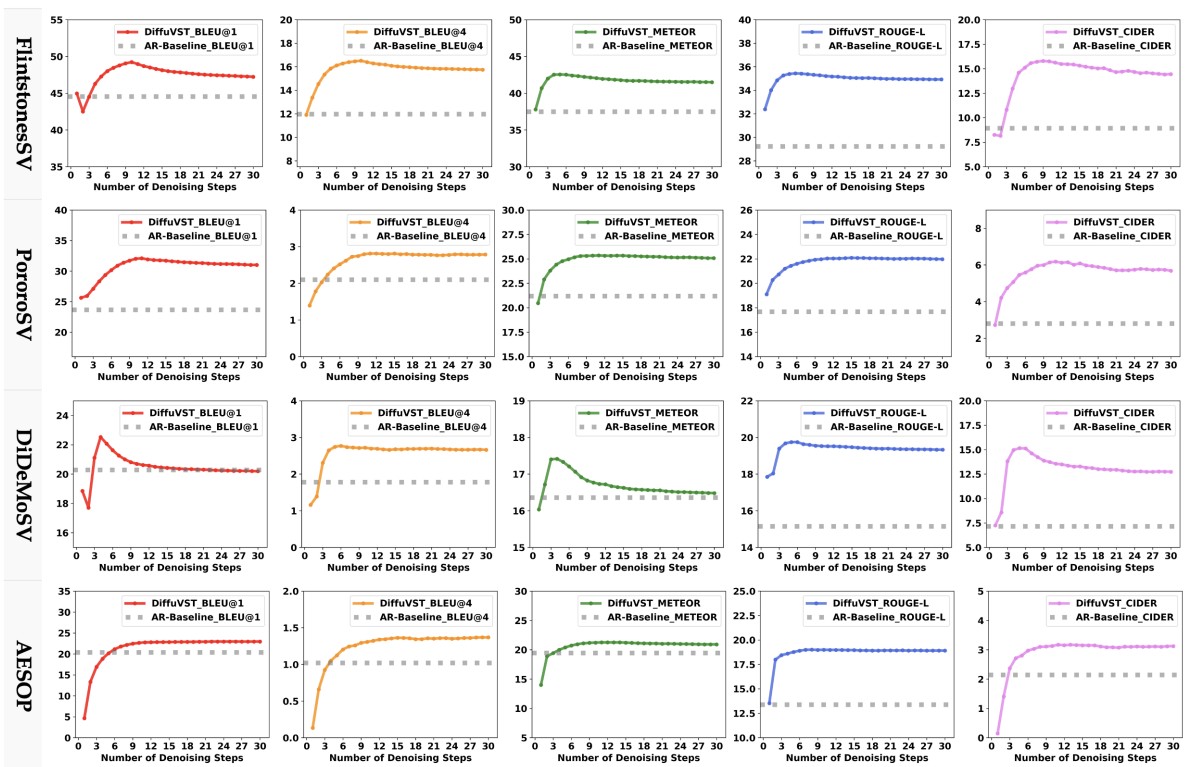

Figure 4: Change of NLG evaluation results of DIFFUVST $_{guided}$ with respect to the number of denoising steps on four datasets. The gray line stands for the performance of GLACNet as a representative Autoregressive(AR)-Baseline. Our DIFFUVST consistently manages to outperform the baseline and reach a stable performance within very few denoising steps, thus significantly reducing inference time.

tently underperforms the fully-equipped version. Therefore, these adapter modules, designed to transfer encoded features to the current domain while also enabling more learnable parameters for aligned feature representation, are indeed a crucial component of our DIFFUVST framework.

## 6 Conclusion

In this paper, we tackle the challenges posed by Visual Storytelling, and present a diffusion-based solution, DIFFUVST that stands out as a novel effort in introducing a non-autoregressive approach to narrating fictional image sequences. Underpinned by its key components, including multimodal encoders with adapter layers, a feature-fusion module and a transformer-based diffusion language model, DIFFUVST models the narration as a single denoising process conditioned on and guided by bidirectional global image- and text- history representations. This leads to generated stories that are not just coherent but also faithfully represent the visual content. Extensive experiments attest the outstanding performance of our non-autoregressive DIFFUVST system both in terms of text generation

quality and inference speed compared to the traditional autoregressive visual storytelling models. Furthermore, we present comprehensive ablation analyses of our method, which demonstrate that our elaborate designs, based on diffusion-related mechanisms, hold great potential for producing expressive and informative descriptions of visual scenes parallelly with a high inference speed.

## Limitations

Our diffusion-based system DIFFUVST inherits a main limitation that resides in the need to carefully search for an optimal hyperparameter setting among various choices and to explain the reason behind its validity. The diffusion process contains numerous key variables that potentially exerts substantial influence on the final model performance - for instance, the strengths and probability of classifier-free guidance as discussed in Section 5.4. Although we are able to deduce a reasonable setup of such parameters via extensive experimental runs, the exact mechanism of how these diffusion-based parameter choices affect our storytelling model remains to be explored.

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

## A  Appendix A. Further Report on Experimental Details

### A.1  Dataset Statistics

As stated in section 5.1, we use four visual-story datasets of synthetic style and content for our experiments: AESOP (Ravi et al., 2021), PororoSV (Li et al., 2018), FlintstonesSV (Maharana and Bansal, 2021) and DiDeMoSV (Maharana et al., 2022). Here are the specific statistics of these datasets: AESOP contains visual stories made of 3 image-text panels, with the visual parts generated by clipart objects. It contains 6,024/991 samples in train/val sets (we use the validation set for testing on AESOP as well). In PororoSV and FlintstonesSV, which contain 10,191/2,334/2,208 and 20,132/2,071/2,309 samples in train/val/test sets respectively, each story sample is a 5-frame image sequence paired

with 5 consecutive texts. DiDeMoSV uses a 3-frame image-text pair sequence as an individual story with a total number of 11,550/2,707/3,378 samples in train/val/test sets.

## A.2 diffuvst implementation parameters

For DIFFUVST's training and inference, we use the following setup: We set the maximum text sequence length to 32 for AESOP/PororoSV/FlintstonesSV (i.e., a maximum story length of $32 * N$ for $N$ images) and 16 for DiDeMoSV which features shorter captions. During training, we randomly sample 30 noised $x_T$ from 1000 diffusion timesteps to form the noisy input embeddings to DIFFUVST. For the strength of textual feature guidance $w$ and the unguided probability $p_{unguide}$, we have explored multiple configurations and leave a detailed discussion of their effects in Section 5.4. During inference, we use random vectors as noisy input $x_T$, and replace all guidance text features with zeros and mask them out. For testing, we denoise a total of 30 steps to iteratively refine the predicted word embeddings and report the best metric performances. We also polish the generated text with unique-consecutive method, where the consecutively repeated words are reduced to only one. We train the whole denoising system with the loss function defined in (7) and the rounding term coefficient ($\lambda$) is set as a dynamic $\lambda$ strategy as stated in Section 4.3. For each dataset, we train DIFFUVST for 30 epochs using AdamW optimizer (Loshchilov and Hutter, 2017) with the initial learning rate of $1e-4$ which is then updated by a Cosine Annealing schedule and a weight decay rate of $1e-2$.