# OpenReview forum: "DiffuVST: Narrating Fictional Scenes with Global-History-Guided Denoising Models"
_EMNLP/2023/Conference — EMNLP 2023 Findings_

### Official Review · Reviewer_Xen1 · 2023-08-01

**Soundness:** 2

**Excitement:**

2: Mediocre: This paper makes marginal contributions (vs non-contemporaneous work), so I would rather not see it in the conference.

**Paper Topic And Main Contributions:**

The paper proposes DiffuVST, a diffusion-based system for visual storytelling that generates a series of visual descriptions as a single conditional denoising process. DiffuVST features a unique design with bi-directional text history guidance and multimodal adapter modules, which improve inter-sentence coherence and image-to-text fidelity. The model outperforms traditional autoregressive models in terms of both text quality and inference speed on four fictional visual-story datasets.
The main contributions of the paper are:
 (1) modeling the visual narration of a set of images as one single denoising process,
(2) proposing global-history guidance and multimodal feature adapter modules, and
(3) demonstrating the effectiveness and practical value of the system.

**Reasons To Accept:**

The paper demonstrates the superiority of DiffuVST over traditional autoregressive models in terms of text quality and inference speed on four fictional visual-story datasets covering both story generation and story continuation tasks.

**Reasons To Reject:**

In section 4.1 an adaptor module for transferring BLIP-encoded multimodal properties to the specific domain of fictional sceneries and texts has been proposed in the study, but its architecture or mathematical formulation has not been specified.
The paper has claimed to a diffusion architecture as the base architecture but the forward training process, backward denoising, and the corresponding scheduler have not been clarified. Is not clear whether a know diffusion model like the Stable DIffuiosn model has been utilized or just a regular LDM module has been used.

The paper does not provide a detailed comparison with other state-of-the-art models in the field of visual storytelling, which makes it difficult to assess the true performance of DiffuVST

A thorough analysis of the limitations of DiffuVST, which could lead to an overestimation of its capabilities.
A clear explanation of the technical details of the proposed model, which could make it difficult for readers to understand and replicate the results.

**Reproducibility:**

2: Would be hard pressed to reproduce the results. The contribution depends on data that are simply not available outside the author's institution or consortium; not enough details are provided.

**Reviewer Confidence:**

3: Pretty sure, but there's a chance I missed something. Although I have a good feel for this area in general, I did not carefully check the paper's details, e.g., the math, experimental design, or novelty.

---

> ### Author Rebuttal · Authors · 2023-08-29
>
> # Responses to Reviewer Xen1
>
> Thank you for such an insightful and constructive review. We truly appreciate your comments and address your concerns and questions as follows:
>
> **[Q1]:**
> Architecture or mathematical formulation of the adapter modules.
>
> **[A1]:**
> Thank you for your inquiry regarding the adapter modules implemented in our work. We appreciate the opportunity to provide a more detailed clarification of its architecture and mathematical formulation, as well as to further demonstrate its significance within our DiffuVST framework:
>
> - **Architecture and mathematical formulation**:
>   Our multimodal adapter modules are implemented as additional linear layers, following the final layers of both the visual and textual encoders. These modules are specifically designed to ensure that the encoded visual and textual feature vectors, $F_v$ and $F_t$, are tailored to the fictional image-text domains (see also Section 4.1). To represent this mathematically, let $V$ denote the output of the BLIP-encoded visual encoder, and $T$ denote the output of the textual encoder. Our adapter layers, represented by matrices $A_v$ and $A_t$, transform these outputs as follows:
>   $$
>   F_v = \sigma(A_v \cdot V + b_v); \\
>   F_t = \sigma(A_t \cdot T + b_t)
>   $$
>   where $\sigma$ denotes the activation function, while $b_v$ and $b_t$ represent the respective bias terms for the visual and textual transformations.
>
> - **Effectiveness and importance of the adapter modules**:
>   To empirically assess the impact of the adapter modules within our DiffuVST system, we conducted ablation studies comparing model performance with and without these adapter layers. We took the "Writer Mode" as the representative task setting and evaluated our models on the four datasets. The results are reported in the table below:
>
>   | Dataset | Method | B@1 | B@4 | M | R | C |
>   |   :----:  | :----:  | :----:  | :----:  | :----:  | :----:  | :----: |
>   | *F.* | DiffuVST- Adapter(+) | **49.76** | **17.01** | **42.98** | **36.22** | **17.63** |
>   | *F.* | DiffuVST- Adapter(-) | 47.6 | 16.17 | 41.65 | 35.41 | 13.09 |
>   | *P.* | DiffuVST- Adapter(+) | **32.09** | **2.81** | **25.35** | **22.03** | **6.18** |
>   | *P.* | DiffuVST- Adapter(-) | 31.01 | 2.58 | 25.06 | 21.75 | 5.15 |
>   | *D.* | DiffuVST- Adapter(+) | **22.08** | **2.74** | **17.33** | **19.75** | **15.16** |
>   | *D.* | DiffuVST- Adapter(-) | 21.1 | 1.72 | 16.93 | 17.62 | 9.19 |
>   | *A.* | DiffuVST- Adapter(+) | **23.13** | **1.41** | **21.14** | **19.07** | **3.4** |
>   | *A.* | DiffuVST- Adapter(-) | 20.52 | 1.28 | 19.75 | 16.84 | 2.42 |
>
>   This table presents the evaluation results of our DiffuVST models with (+) and without (-) adapter layers, all using the same hyperparameter settings. The "(+)Adapter" versions correspond to the "DiffuVST_*guided*" models reported in Tables 1 and 2 of the paper. The abbreviations  *F.*, *P.*, *D.*, and *A.* stand for the datasets FlintstonesSV, PororoSV, DiDeMoSV, and AESOP, respectively. The metrics include BLEU@1, BLEU@4, METEOR, ROUGE-L, and CIDEr. As evident from the table, the inclusion of adapter modules consistently improves model performance **across all evaluation metrics and datasets**. Models built without adapters consistently underperform the fully-equipped version. Therefore, these adapter modules, designed to adapt encoded features to the target domain while enabling more learnable parameters for aligned feature representation, are a vital component of our DiffuVST framework.
>
> We sincerely appreciate your question, and we are committed to providing a more detailed formal definition of the adapters and incorporating the ablation studies on adapter modules into the main text of our revised paper. This will ensure that the significance of adapter modules, as demonstrated in the table and our discussion, receives appropriate emphasis in the paper.
>
>
> **[Q2]:**
> More detailed comparison with other state-of-the-art models.
>
> **[A2]:** We appreciate the reviewer's constructive feedback regarding the need for comparing our DiffuVST with more baselines to validate its effectiveness. We acknowledge the importance of such comparisons in the context of more recent advancements in the field. To address this concern, we have incorporated additional state-of-the-art open-source auto-regressive baselines in order to strengthen the validation of our approach:
>
> 1. **TAPM [1, CVPR2021]:**
> TAPM aligns the embedding space of a pretrained visual encoder and language generator using and Adaptation Loss. Subsequently, it applies a Sequential Coherence Loss for finetuning, achieving superior VIST results over multiple visual storytelling baselines including AREL[4], INet[5], HSRL[6], etc. For our task, we utilize a pretrained ViT as the visual encoder and GPT2 for the language generator, ensuring a fair comparison, as our DiffuVST also employs a ViT encoder (BLIP-Image-Model) for image features.
>
> 1. **BLIP-EncDec [2, ICML2022]:**
> This baseline leverages both the Encoder and LM-Decoder of the pretrained BLIP captioning model, fine-tuned on the image-text pairs from our datasets. During inference, we incorporate generated captions from previous images in the textual history of the story sequence, introducing a sequential dependency for captioning the current image. Given that BLIP-EncDec builds upon a well-established captioning model, it serves as a strong baseline, with the mentioned modification specifically for visual storytelling.
>
> Considering the **accessibility** and **reproducibility** of visual storytelling models, we believe these two strong baselines, along with GLACNet[3], which was part of our initial submission, provide a comprehensive quantitative comparison for our model.
>
> The results of this extended evaluation are presented in the table below, which supplements Table 1 and 2 from our initial submission. We use "Writer Mode" as an example task setting and "DiffuVST" refers to the optimally guided version of our model, denoted as "DiffuVST_*guided*" in Table 1 and 2.
>
> | Dataset | Method | B@1 | B@4 | M | R | C | Inference_Time(sec.) ↓ |
> |   :----:  | :----:  | :----:  | :----:  | :----:  | :----:  | :----: | :----: |
> | *F.* | GLACNet [3] | 44.55 | 11.98 | 37.48 | 29.23 | 8.93 | 1.2042 |
> | *F.* | BLIP-EncDec [2] | 48.75 | 16.32 | 42.25 | 35.3 | 15.94 | 2.0486 |
> | *F.* | TAPM [1] | 49.05 | 15.08 | 42.12 | 33.03 | 16.54 | 1.8923 |
> | *F.* | DiffuVST (ours) | **49.76** | **17.01** | **42.98** | **36.22** | **17.63** | **0.0981** |
> |     |   |   |   |   |   |  |  |
> | *P.* | GLACNet [3] | 23.66 | 2.1 | 21.19 | 17.67 | 2.8 | 0.7844 |
> | *P.* | BLIP-EncDec [2] | 31.38 | 2.96 | 25.13 | 21.64 | 5.63 | 1.8702 |
> | *P.* | TAPM [1] | 28.43 | **3.78** | 24.43 | 21.87 | 5.55 | 1.2289 |
> | *P.* | DiffuVST (ours) | **32.09** | 2.81 | **25.35** | **22.03** | **6.18** | **0.1196** |
> |     |   |   |   |   |   |  |  |
> | *D.* | GLACNet [3] | 20.28 | 1.78 | 16.36 | 15.15 | 7.15 | 0.3030 |
> | *D.* | BLIP-EncDec [2] | 20.02 | 2.14 | 16.01 | 17.74 | 14.7 | 0.7474 |
> | *D.* | TAPM [1] | 20.79 | 2.12 | 16.35 | 17.68 | **15.26** | 0.4457 |
> | *D.* | DiffuVST (ours) | **22.08** | **2.74** | **17.33** | **19.75** | 15.16 | **0.0161** |
> |     |   |   |   |   |   |  |  |
> | *A.* | GLACNet [3] | 20.37 | 1.02 | 19.46 | 13.39 | 2.14 | 0.9285 |
> | *A.* | BLIP-EncDec [2] | 19.16 | **1.81** | 21.88 | 16.71 | 2.89 | 1.3141 |
> | *A.* | TAPM [1] | 21.95 | 1.56 | 21.55 | 16.3 | 2.97 | 1.1629 |
> | *A.* | DiffuVST (ours) | **23.13** | 1.41 | **21.14** | **19.07** | **3.4** | **0.1988** |
>
> As evident from the table, DiffuVST consistently outperforms these baselines **across nearly all metrics and datasets**. While there are marginal exceptions, such as the slightly lower B@4 score on the PororoSV and ASEOP datasets and the CIDEr score on DiDeMoSV compared to TAPM, these do not diminish DiffuVST's overall superior performance. Notably, given TAPM has outperformed most state-of-the-art visual storytellers in their results on VIST (as indicated in the TAPM paper[1]), we have good reason to believe that our model can also surpass these models, further demonstrating the effectiveness of DiffuVST. Moreover, it is essential to emphasize that DiffuVST's non-autoregressive nature significantly accelerates inference, making it **more than 10x times faster** than the auto-regressive baselines. This acceleration greatly enhances the efficiency of visual storytelling. In conclusion, DiffuVST not only achieves top-tier performance but also excels in efficiency, making it an overall successful choice for the task of visual storytelling.
>
> Once again, we sincerely thank the reviewer for prompting this comprehensive evaluation, and we will incorporate these additional baselines into our revised paper, further enhancing its comprehensiveness and validation of DiffuVST's capabilities.
>
>
> **[Q3]:**
> A more thorough analysis of limitations.
>
> **[A3]:**
> We sincerely appreciate your feedback and the opportunity to delve deeper into the limitations of our DiffuVST system. As mentioned in the "Limitations" section (Lines 562-575) of our paper, we acknowledge a primary limitation revolving around the need for meticulous hyperparameter search. Allow us to provide a more extensive explanation of how these hyperparameter choices affect the performance of our diffusion-based visual storyteller. Specifically, we will focus on two critical hyperparameters:
>
> 1. **Textual feature gidance parameters (unguided probability $p_{unguided}$ and strength $w$)**:
>     In our system's textual feature guidance mechanism (see Section 4.2, Lines 291-321, Eq6), $p_{unguided}$ regulates the probability of training the model solely based on visual data, without any textual cues (i.e., not guided by global textual history), and $w$ controls the weighting of the guided model. The various configurations of these two key hyperparameters significantly impact our model's performance.
>    As detailed in Section 5.3, using the DiDeMoSV dataset as an example, we experimented with different values of guidance strength $w\in\{0.0, 0.3, 0.5, 0.7, 1.0\}$ and unguided probability $p_{unguided}\in\{0.3, 0.5, 0.7\}$. Only after conducting these exhaustive experiments did we determine that the optimal settings for training our DiffuVST model were a strong guidance weight ($w=1.0$) and a relatively low guidance rate (i.e., a high unguided probability of $ p_{unguided}=0.7$) (see also Table 3 and Figure 3).
>
> 2. **Rounding term coefficient $\lambda$**:
>     As defined in Section 4.3 (Lines 352-364) and Eq7, the rounding term coefficient $\lambda$, which regulates the relative weight of $L^\prime$ and $L_R$, is another crucial hyperparameter in our model's training. In our preliminary experiments with fixed $\lambda$ values (ranging from 0.1 to 0.9), we observed that $L^\prime$ and $L_R$ diverged in scale during training, regardless of the chosen $\lambda$. This divergence resulted in an undesirable situation where the relative values of $L^\prime$ or $L_R$ became either too high or too low, destabilizing the training process Consequently, we had to adopt a dynamic $\lambda$ strategy (see Lines 359-364) that allowed both $L^\prime$ and $L_R$ to contribute equally to gradient updates, depending on the respective status of each training step. This dynamic $\lambda$ strategy proved to be more effective in maintaining training stability.
>     To further underscore the superiority of the dynamic $\lambda$ strategy, we conducted an experimental comparison between dynamic $\lambda$ and a fixed $\lambda$ configuration ($\lambda=0.3$, which yielded the most stable training loss curve and model performance among fixed $\lambda$ choices), as shown in the table below:
>
>     | Dataset | Method | B@1 | B@4 | M | R | C |
>     |   :----:  | :----:  | :----:  | :----:  | :----:  | :----:  | :----: |
>     | *F.* | DiffuVST- λ=*dynamic* | **49.76** | **17.01** | **42.98** | **36.22** | **17.63** |
>     | *F.* | DiffuVST- λ=*0.3* | 48.91 | 16.35 | 41.54 | 35.48 | 10.99 |
>     | *P.* | DiffuVST- λ=*dynamic* | **32.09** | 2.81 | **25.35** | **22.03** | **6.18** |
>     | *P.* | DiffuVST- λ=*0.3* | 31.24 | **2.95** | 25.15 | 22 | 5.25 |
>     | *D.* | DiffuVST- λ=*dynamic* | **22.08** | **2.74** | **17.33** | **19.75** | **15.16** |
>     | *D.* | DiffuVST- λ=*0.3* | 21.35 | 2.45 | 16.7 | 19.44 | 13.02 |
>     | *A.* | DiffuVST- λ=*dynamic* | **23.13** | **1.41** | **21.14** | **19.07** | **3.4** |
>     | *A.* | DiffuVST- λ=*0.3* | 21.13 | 1.31 | 20.7 | 16.94 | 2.81 |
>
>     This table presents the evaluation results in "Writer Mode" for our DiffuVST model trained with both dynamic $\lambda$ and fixed $\lambda=0.3$ configurations, using identical hyperparameter settings. Notably, the dynamic $\lambda$ strategy consistently enhances the performance of DiffuVST **across all evaluation metrics and datasets**, except for a minor deviation (B@4 on the PororoSV dataset).
>
> Once again, we sincerely appreciate the opportunity to discuss the limitations of DiffuVST in greater detail. We plan to incorporate this comprehensive limitation analysis into the appendices of the revised version of our paper, as it promises to inspire future research endeavors.
>
>
> **[Q4]:**
> A clearer explanation of the technical details including clarification on the diffusion model and process implemented in our work.
>
> **[A4]:**
> Thank you for your valuable feedback regarding the need for a comprehensive explanation of the technical details involved in our proposed method. We fully recognize the importance of clarity, especially for the purpose of reproducibility. To address this concern, we are committed to providing a thorough and accessible account of the technical details behind our implementation:
>
> 1. **Diffusion model and process details**:
>    Our DiffuVST model is constructed upon the foundation of the Diffusion-LM architecture, incorporating both forward process for adding noise and the backward processes for denoising:
>
>    **The forward process** initiates with a ground truth word embedding $x_0$, and progressively introduces noise over $T$ steps (following a noise scheduler, to be discussed in the next paragraph) to produce $x_T$, utilizing a standard cosine noise scheduler. Subsequently, the corrupted embedding, $x_T$, is subjected to **the backward denoising process**, along with the adapter-tuned visual and textual features as guidance, undergoes the backward denoising process, where denoising transformer directly restores the original textual embedding $x_0$.
>
>    **The noise schedule**, parametrized by $\bar{a}_{1:T}$, serves as an important hyperparameter in the Diffusion-LM model shared across all diffusion steps. In our work, we employed the standard **cosine noise schedule** in line with the strategy taken in Improved DDPM[7]. The underlying mechanism could be expressed as follows (see also Line 187), where $q$ denotes a forward noise injection process, which introduces Gaussian noise at time t with variance $β_t ∈ (0, 1)$ based on the $\bar{a}$ cosine schedule:
>
>     $$
>       q(x_t|x_{t-1}) = N(x_t; \sqrt{1-\beta_t}x_{t-1},\beta_tI),
>     $$
>
>     $$
>       \beta_t=1-\frac{\bar{a}_t}{\bar{a}_{t-1}}
>     $$
>
>     $$
>       \bar{a}_t = \frac{f(t)}{f(0)}
>     $$
>
>     $$
>       f(t) = \cos{(\frac{t/T+s}{1+s} \cdot \frac{\pi}{2})^2}
>     $$
>
> 2. **Other implementation details**:
>
>     **Model backbones**: The textual and visual feature extractors within our DiffuVST are implemented as the text and image encoder of the pretrained BLIP[2] model. The denoising transformer leverages the architecture of DISTILBERT-BASE-UNCASED[8].
>
>     **Hyperparameters in training and inference:** All hyperparameters associated with training and inference are available in Appendix A.2 (Line 766-798). Several key parameters warrant emphasis here: We configure the total number of diffusion steps to 1000. The text sequence length is set to 32 for AESOP/PororoSV/FlintstonesSV and 16 for DiDeMoSV. We train DiffuVST for 30 epochs, employing the AdamW optimizer with an initial learning rate of $1e-4$, subsequently adjusted using a Cosine Annealing schedule a weight decay rate of $1e−2$.
>
> Furthermore, we would like to ensure that we are committed to sharing our codebase and dataset publicly to facilitate further research and validation of our method. Once again, thank you for highlighting this clarity issue.
>
>
>
> **[References]:**
> [1] Integrating Visuospatial, Linguistic, and Commonsense Structure into Story Visualization. EMNLP 2021.
> [2] BLIP: Bootstrapping Language-Image Pre-training for Unified Vision-Language Understanding and Generation. ICML 2022.
> [3] GLAC Net: GLocal Attention Cascading Networks for Multi-image Cued Story Generation. NAACL 2018.
> [4] No Metrics Are Perfect: Adversarial Reward Learning for Visual Storytelling. ACL 2018.
> [5] Hide-and-Tell: Learning to Bridge Photo Streams for Visual Storytelling. AAAI 2020.
> [6] Hierarchically Structured Reinforcement Learning for Topically Coherent Visual Story Generation. AAAI 2019.
> [7] Improved Denoising Diffusion Probabilistic Models. 2021.
> [8] DistilBERT, a distilled version of BERT: smaller, faster, cheaper and lighter. NeurIPS 2019.

---

### Official Review · Reviewer_sjRb · 2023-08-01

**Soundness:** 4

**Excitement:**

4: Strong: This paper deepens the understanding of some phenomenon or lowers the barriers to an existing research direction.

**Paper Topic And Main Contributions:**

The paper proposes a diffusion model (DiffuVST) to generate sentence captions for describing fictional scenes. The DiffuVST uses a multimodal adapter module for image and text encoding, and uses text history guidance in the diffusion process. Experimental results show that the DiffuVST achieves better captioning performance and also has faster inference speed.

**Questions For The Authors:**

Questions about the method:
1. The eq7 is less clear. N should be T, right? Why does L339 say “we add a x1 restoring loss to eq (4)”?  Is this loss already included in eq4? Why do you use L1 loss instead of L2?
2. How important is the dynamic \lambda strategy in L360-364?

Questions about the results:
1. Why can the text guidance improve the performance in the writer mode since the texts are masked as zeros?
2. Why do more diffusion steps result in worse performance as shown in Figure 4?

**Reasons To Accept:**

- The paper is well-written and easy to follow.

- The DiffuVST model is the first to leverage diffusion models in visual storytelling. It also explores the use of global-history guidance to improve coherence and visual fidelity.

- Experiments on four datasets demonstrate the effectiveness of the approach with better quantitative performance and higher inference speed.

**Reasons To Reject:**

- The multimodal encoder contains less novelty.
The paper claims the adapter modules in multimodal encoders as one of the major contributions. However, they are just linear layers to transform the pretrained features, which have been used widely in the literature. Besides, the effectiveness of the adapter module has not been verified in the experiments.

- More baselines should be compared to validate the effectiveness of the diffusion model.
The paper only utilizes a LSTM based model as a baseline to compare the auto-regressive and non-autoregressive diffusion models. More state-of-the-art auto-regressive models should be compared. It would be easy to adapt existing image or video captioning models for visual storytelling. For example, one can replace the decoder in DiffuVST by a transformer-based auto-regressive decoder and keep the same encoder for comparison.

- No qualitative results are provided to analyze the success and failure cases.
More analysis should be provided to show the advantages of the diffusion model besides the inference speed, especially considering that the model is sensitive to hyperparameters.
Human evaluations are encouraged since the task is subjective.

**Reproducibility:**

3: Could reproduce the results with some difficulty. The settings of parameters are underspecified or subjectively determined; the training/evaluation data are not widely available.

**Reviewer Confidence:**

4: Quite sure. I tried to check the important points carefully. It's unlikely, though conceivable, that I missed something that should affect my ratings.

**Typos Grammar Style And Presentation Improvements:**

Table 1 and table 2 can be merged together to save space. It would also be good to add citations for the compared baselines.

Figure 2 is not very clear. It is hard to understand how the framework works such as the inputs, outputs and the diffusion process.

---

> ### Author Rebuttal · Authors · 2023-08-29
>
> # Responses to Reviewer sjRb
>
> Thank you for such an insightful and constructive review. We truly appreciate your comments and address your concerns and questions as follows:
>
> **[Q1]:**
> Motivations and effectiveness of the adapter modules.
>
> **[A1]:**
> We appreciate your concern regarding our motivations behind highlighting the adapter modules as well as the experimental verification of their effectiveness. In response to this, we would like to provide the following points to address your concern:
>
> 1. **Motivations for Introducing Adapter Modules:**
>    - ***Domain adaptation:*** Firstly, the adapter modules serve a crucial purpose in bridging the gap between general-domain pretrained features and the specific fictional image-text domain, as you precisely pointed out. These modules facilitate domain transfer, enabling our model to adapt and perform effectively in fictional scenes.
>    - ***Enhancing feature alignment:*** Additionally, the image and text features are utilized to provide guidance for the optimization of our DiffusionLM-based decoder. By incorporating adapter modules, we introduce additional trainable parameters that better align the features extracted by the encoders with the decoding capabilities of the DiffusionLM, thereby enhancing the overall framework integrity.
>
> 2. **Effectiveness of the adapter modules:**
>     To empirically validate the significance of the adapter modules within our DiffuVST system, we have conducted ablation studies where we compared the performance with and without the adapter layers. We took the "Writer Mode" as the representative task setting and evaluated our models on the four datasets. The results are presented in the table below:
>
>     | Dataset | Method | B@1 | B@4 | M | R | C |
>     |   :----:  | :----:  | :----:  | :----:  | :----:  | :----:  | :----: |
>     | *F.* | DiffuVST- Adapter(+) | **49.76** | **17.01** | **42.98** | **36.22** | **17.63** |
>     | *F.* | DiffuVST- Adapter(-) | 47.6 | 16.17 | 41.65 | 35.41 | 13.09 |
>     | *P.* | DiffuVST- Adapter(+) | **32.09** | **2.81** | **25.35** | **22.03** | **6.18** |
>     | *P.* | DiffuVST- Adapter(-) | 31.01 | 2.58 | 25.06 | 21.75 | 5.15 |
>     | *D.* | DiffuVST- Adapter(+) | **22.08** | **2.74** | **17.33** | **19.75** | **15.16** |
>     | *D.* | DiffuVST- Adapter(-) | 21.1 | 1.72 | 16.93 | 17.62 | 9.19 |
>     | *A.* | DiffuVST- Adapter(+) | **23.13** | **1.41** | **21.14** | **19.07** | **3.4** |
>     | *A.* | DiffuVST- Adapter(-) | 20.52 | 1.28 | 19.75 | 16.84 | 2.42 |
>
>     This table presents the evaluation results of our DiffuVST with (+) and without (-) Adapter layers, all under the same hyperparameter settings, including $\lambda$, guidance strength, and guidance probability. The "(+)Adapter" versions are thus identical to "DiffuVST_guided" reported in Table 1 & 2 in the paper. The abbreviations *F.*, *P.*, *D.*, and *A.* correspond to the datasets FlintstonesSV, PororoSV, DiDeMoSV, and AESOP, respectively. The metrics include BLEU@1, BLEU@4, METEOR, ROUGE-L and CIDEr. As vividly demonstrated in the above table, the presence of adapter modules significantly impacts model performance **across all evaluation metrics and datasets**, where DiffuVST built without adapters consistently underperforms the fully-equipped version. Therefore, these adapter modules, designed to transfer encoded features to the current domain while also enabling more learnable parameters for aligned feature representation, are indeed a crucial component of our DiffuVST framework.
>
> We regret that, due to the space constraints in our initial submission, we had to prioritize a detailed discussion on the "text feature guidance" mechanism (Section 5.3) over the ablation studies on adapter layers. However, thanks to your insightful suggestion that Table 1 & 2 can be merged to save space (see **[Q10]**), we will incorporate the critical ablation studies on adapter modules into the main text in our revised paper. This will ensure that the importance of adapter modules is appropriately highlighted, considering their substantial impact on our DiffuVST system's performance, as demonstrated in the above table and discussion.
>
> **[Q2]:**
> More baselines should be compared to validate the effectiveness of our DiffuVST.
>
> **[A2]:**
> We appreciate the reviewer's constructive feedback regarding the need for comparing our DiffuVST with more baselines to validate its effectiveness. We acknowledge the importance of such comparisons in the context of more recent advancements in the field. To address this concern, we have incorporated additional state-of-the-art open-source auto-regressive baselines in order to strengthen the validation of our approach:
>
> 1. **TAPM [1, CVPR2021]:**
> TAPM aligns the embedding space of a pretrained visual encoder and language generator using and Adaptation Loss. Subsequently, it applies a Sequential Coherence Loss for finetuning, achieving superior VIST results over multiple visual storytelling baselines including AREL[4], INet[5], HSRL[6], etc. For our task, we utilize a pretrained ViT as the visual encoder and GPT2 for the language generator, ensuring a fair comparison, as our DiffuVST also employs a ViT encoder (BLIP-Image-Model) for image features.
>
> 1. **BLIP-EncDec [2, ICML2022]:**
> This baseline leverages both the Encoder and LM-Decoder of the pretrained BLIP captioning model, fine-tuned on the image-text pairs from our datasets. During inference, we incorporate generated captions from previous images in the textual history of the story sequence, introducing a sequential dependency for captioning the current image. Given that BLIP-EncDec builds upon a well-established captioning model, it serves as a strong baseline, with the mentioned modification specifically for visual storytelling.
>
> Considering the **accessibility** and **reproducibility** of visual storytelling models, we believe these two strong baselines, along with GLACNet[3], which was part of our initial submission, provide a comprehensive quantitative comparison for our model.
>
> The results of this extended evaluation are presented in the table below, which supplements Table 1 and 2 from our initial submission. We use "Writer Mode" as an example task setting and "DiffuVST" refers to the optimally guided version of our model, denoted as "DiffuVST_*guided*" in Table 1 and 2.
>
> | Dataset | Method | B@1 | B@4 | M | R | C | Inference_Time(sec.) ↓ |
> |   :----:  | :----:  | :----:  | :----:  | :----:  | :----:  | :----: | :----: |
> | *F.* | GLACNet [3] | 44.55 | 11.98 | 37.48 | 29.23 | 8.93 | 1.2042 |
> | *F.* | BLIP-EncDec [2] | 48.75 | 16.32 | 42.25 | 35.3 | 15.94 | 2.0486 |
> | *F.* | TAPM [1] | 49.05 | 15.08 | 42.12 | 33.03 | 16.54 | 1.8923 |
> | *F.* | DiffuVST (ours) | **49.76** | **17.01** | **42.98** | **36.22** | **17.63** | **0.0981** |
> |     |   |   |   |   |   |  |  |
> | *P.* | GLACNet [3] | 23.66 | 2.1 | 21.19 | 17.67 | 2.8 | 0.7844 |
> | *P.* | BLIP-EncDec [2] | 31.38 | 2.96 | 25.13 | 21.64 | 5.63 | 1.8702 |
> | *P.* | TAPM [1] | 28.43 | **3.78** | 24.43 | 21.87 | 5.55 | 1.2289 |
> | *P.* | DiffuVST (ours) | **32.09** | 2.81 | **25.35** | **22.03** | **6.18** | **0.1196** |
> |     |   |   |   |   |   |  |  |
> | *D.* | GLACNet [3] | 20.28 | 1.78 | 16.36 | 15.15 | 7.15 | 0.3030 |
> | *D.* | BLIP-EncDec [2] | 20.02 | 2.14 | 16.01 | 17.74 | 14.7 | 0.7474 |
> | *D.* | TAPM [1] | 20.79 | 2.12 | 16.35 | 17.68 | **15.26** | 0.4457 |
> | *D.* | DiffuVST (ours) | **22.08** | **2.74** | **17.33** | **19.75** | 15.16 | **0.0161** |
> |     |   |   |   |   |   |  |  |
> | *A.* | GLACNet [3] | 20.37 | 1.02 | 19.46 | 13.39 | 2.14 | 0.9285 |
> | *A.* | BLIP-EncDec [2] | 19.16 | **1.81** | 21.88 | 16.71 | 2.89 | 1.3141 |
> | *A.* | TAPM [1] | 21.95 | 1.56 | 21.55 | 16.3 | 2.97 | 1.1629 |
> | *A.* | DiffuVST (ours) | **23.13** | 1.41 | **21.14** | **19.07** | **3.4** | **0.1988** |
>
> As evident from the table, DiffuVST consistently outperforms these baselines **across nearly all metrics and datasets**. While there are marginal exceptions, such as the slightly lower B@4 score on the PororoSV and ASEOP datasets and the CIDEr score on DiDeMoSV compared to TAPM, these do not diminish DiffuVST's overall superior performance. Notably, given TAPM has outperformed most state-of-the-art visual storytellers in their results on VIST (as indicated in the TAPM paper[1]), we have good reason to believe that our model can also surpass these models, further demonstrating the effectiveness of DiffuVST. Moreover, it is essential to emphasize that DiffuVST's non-autoregressive nature significantly accelerates inference, making it **more than 10x times faster** than the auto-regressive baselines. This acceleration greatly enhances the efficiency of visual storytelling. In conclusion, DiffuVST not only achieves top-tier performance but also excels in efficiency, making it an overall successful choice for the task of visual storytelling.
>
> Once again, we sincerely thank the reviewer for prompting this comprehensive evaluation, and we will incorporate these additional baselines into our revised paper, further enhancing its comprehensiveness and validation of DiffuVST's capabilities.
>
>
>
> **[Q3]:**
> Qualitative results, analysis of success/fail cases, and human evaluation.
>
> **[A3]:**
> Thank you for raising important points regarding the need for qualitative analysis and possibly human evaluation as well. We appreciate your feedback and would like to address these aspects as follows:
>
> - **Qualitative result comparison**: Expanding our evaluation beyond quantitative metrics surely allows us to delve into nuanced detilas in terms of e.g., text-image alignment and story coherence. To address this, we have selected several visual stories generated by all compared models. In our revised paper, we will present these examples and provide a detailed qualitative analysis, offering insights into the difference in quality of the generated content by compared models.
> - **Human Evaluation**: We fully recognize the importance of human judgment when assessing the quality of narrative generation, considering the subjective nature of storytelling. In response to your valuable suggestion, we are currently conducting a user study to assess the stories generated by our model and the baselines. Our aim is to incorporate a detailed user study in our revised paper or its appendices, which will provide valuable insights from human perspectives, enriching the assessment of our model's capabilities.
>
> We sincerely appreciate your advice that help guide us towards a more comprehensive evaluation of our work, and we are committed to addressing these qualitative aspects of evaluation to enhance the integrity of our paper.
>
>
> **[Q4]:**
> More clarification on Eq7, $N$ or $T$?
>
> **[A4]:**
> Thank you for your insightful question concerning Eq7. We appreciate the opportunity to clarify the notations $N$ v.s. $T$ and their underlying concepts. In Eq7, the term $L^\prime$ indeed employs a different set of diffusion time steps compared to the original $L$ in Eq4. We explain the differences as follows:
>
> - The $L$ term in Eq4 originates from DDPM [7], serving as the foundational loss function for all diffusion-based systems. As initially conceived in DDPM, $T$ should be a large number, such as 1000, for adding noise (refer also to Line 190-191). This choice aligns with the reverse process (denoising), which then necessitates significantly large steps, e.g., 500. Unfortunately, this DDPM approach results in slow inference, which contradicts our intention for a diffusion-based visual storyteller.
>
> - To accelerate the process, we actually employed the strategy inspired by Improved DDPM [8], as implemented in DiffusionLM [9]. Here, we **sample a subset of size $N$ (e.g., 100) from the total $T$ timesteps** to serve as the actual set of diffusion time steps:
>   $$
>   S = \{s_0, ..., s_N | s_t < s_{t−1} \} \in (0,T]
>   $$
>   Our LM is trained to predict $x_0$ directly from $x_t$, where $t\in(0, N]$, sampled from this subset $S$. Consequently, our model can generate high-quality output within only a few denoising steps, as evidenced in Section 5.2 and Figure 4. This approach significantly reduces inference time.
>
> - In short, the $L^\prime$ term in Eq7 indeed uses a different set of diffusion time steps ($N$) compared to the original DDPM's $L$ term in Eq4 ($T$). These smaller steps in $L^\prime$ are designed to accelerate the denoising process to match our goal for DiffuVST.
>
> Again, we sincerely appreciate your question about this detail and will ensure that future versions of our paper include clearer definitions and necessary explanations of our equations to eliminate any potential confusion.
>
>
> **[Q5]:**
> The $x_1$ restoring loss (Line 339) in relation to Eq4 & Eq7.
>
> **[A5]:**
> We highly appreciate your thoughtful question regarding the "$x_1$ restoring loss" and its relation to Eq4 and Eq7 in our paper. Allow us to provide here a clarification:
>
> - The "$x_1$ restoring loss" is an addition to the usual "$x_t$ restoring loss" as implemented in Improved DDPM [8] and DiffusionLM [9]. It serves to regulate the training process of our model, which learns not only to predict $x_0$ from a randomly sampled diffusion timestep $t \in N$ (see **[A4]** to **[Q4]**), but also consistently from the timestep $t_1$.
>
> - Thus, the "$x_1$ restoring loss" is not part of the original DDPM loss function in Eq4. But our proposed final loss function Eq7 does include this "$x_1$ restoring loss" in the term $L^\prime$, as stated in Line 345-346. Here, we revise our final loss function Eq7:
>     $$
>     L_{final}=L^\prime + \lambda L_R
>     =\sum_{t=1}^{N}E_{q(x_{t}|x_0)}[\left\|\mu_\theta(x_{t}, t)-\hat{\mu}(x_{t},x_0)\right\| + \left\|\mu_\theta(x_1,1)-\hat{\mu}(x_1,x_0)\right\| -\lambda\log{p_\theta(w|\hat{x})}]
>     $$
>
>     It comprises of "$x_t$ restoring loss" ($x_{0:t}$), "$x_1$ restoring loss" ($x_{0:1}$) and a rounding term ($L_R$).
>
> We regret that we missed the "$x_1$ restoring loss" term in Eq7 of our initial submission. And we sincerely thank you for identifying this oversight. We will rectify this in the revised version of our paper and ensure better accuracy and completeness of the content.
>
>
> **[Q6]:**
> The reasons behind using L1 instead of L2 for $L^\prime$ ($x_t$ and $x_1$ restoring loss) in Eq7.
>
> **[A6]:**
> Thank you for your insightful question regarding our choice of using L1 instead of L2 for the $x_t$ and $x_1$ restoring loss, denoted as $L^\prime$ in Eq7. Our decision was motivated by both observations and theoretical considerations:
>
> - **Observations:** During our preliminary experiments, we noticed that employing the original L2 formulation for $L^\prime$ often resulted in rather turbulent training loss curves. In contrast, when we utilized the L1 distance metric, we observed **more consistent and stable loss convergence**. This led us to favor L1 as a means to enhance training stability.
>
> - **Theoretical Rationale:** We hypothesize that the improved stability with L1 can be attributed to its lower sensitivity to outliers, a crucial aspect given the diverse hyperparameter configurations inherent in our diffusion-based system. Consequently, substituting L2 with L1 appears to promote a **more stable training process**. This choice also aligns with findings in related literature, such as [10], which reported similar advantages associated with L1 loss.
>
> Once again, we greatly appreciate your question about our choice of L1 over L2. We will make sure to state clearly the motivations behind this decision in our revised paper to avoid confusion.
>
>
> **[Q7]:**
> Importance of the dynamic $\lambda$ strategy (Line 360-364).
>
> **[A7]:**
> We appreciate your question regarding the significance of the dynamic $\lambda$ strategy implemented by our work. We would like to highlight its importance from both observational and empirical perspectives:
>
> - **Observations:** $\lambda$, the rounding term coefficient, serves to regulate the relative weight of $L^\prime$ and $L_R$ in Eq7, so that both loss terms should decrease at a proper pace. In our preliminary experiments with fixed $\lambda$ values (ranging from 0.1 to 0.9), we observed that $L^\prime$ and $L_R$ inevitably diverged in scale during training, regardless of the chosen $\lambda$. This divergence led to an undesirable situation, where the relative values of $L^\prime$ or $L_R$ became either too high or too low, undermining training stability. To maintain a more stable training process, we were thus motivated to adopt a dynamic $\lambda$ setting, enabling both $L^\prime$ and $L_R$ to contribute equally to gradient updates, according to the respective status of each training step.
>
> - **Empricial evidence:** In the fixed $\lambda$ setting, we identified that $\lambda=0.3$ yielded the most stable training loss curve and model performance. To assess the impact of dynamic $\lambda$, we have conducted a comparison between dynamic $\lambda$ and the fixed $\lambda=0.3$ configuration in terms of model performance. Similar to **[A1]**, we use the "Writer Mode" as the representative task setting and present the evaluation results on the four datasets in the table below:
>
>     | Dataset | Method | B@1 | B@4 | M | R | C |
>     |   :----:  | :----:  | :----:  | :----:  | :----:  | :----:  | :----: |
>     | *F.* | DiffuVST- λ=*dynamic* | **49.76** | **17.01** | **42.98** | **36.22** | **17.63** |
>     | *F.* | DiffuVST- λ=*0.3* | 48.91 | 16.35 | 41.54 | 35.48 | 10.99 |
>     | *P.* | DiffuVST- λ=*dynamic* | **32.09** | 2.81 | **25.35** | **22.03** | **6.18** |
>     | *P.* | DiffuVST- λ=*0.3* | 31.24 | **2.95** | 25.15 | 22 | 5.25 |
>     | *D.* | DiffuVST- λ=*dynamic* | **22.08** | **2.74** | **17.33** | **19.75** | **15.16** |
>     | *D.* | DiffuVST- λ=*0.3* | 21.35 | 2.45 | 16.7 | 19.44 | 13.02 |
>     | *A.* | DiffuVST- λ=*dynamic* | **23.13** | **1.41** | **21.14** | **19.07** | **3.4** |
>     | *A.* | DiffuVST- λ=*0.3* | 21.13 | 1.31 | 20.7 | 16.94 | 2.81 |
>
>     This table shows the evaluation results of our DiffuVST model trained with both dynamic $\lambda$ and fixed $\lambda=0.3$ configurations, using identical hyperparameter settings. The "$\lambda=dynamic$" versions correspond to "DiffuVST_*guided*" as reported in Table 1 & 2 in the paper. Notably, the dynamic $\lambda$ strategy consistently enhances the performance of DiffuVST **across all evaluation metrics and datasets**, except for a minor deviation (B@4 on PororoSV dataset). This evidence clearly underscores the significant role of dynamic $\lambda$ in optimizing our DiffuVST model's loss function.
>
> We are sincerely grateful for the opportunity to elaborate on how the dynamic $\lambda$ strategy impacts our framework. We also plan to emphasize this discussion in the revised version of our paper, given its clear influence on model performance, as demonstrated in the table above.
>
>
>
> **[Q8]:**
> Why does the text guidance improve model performance even if ground-truth texts are masked during inference?
>
>
> **[A8]:**
> We thank the reviewer for raising this insightful question regarding the mechanism of textual feature guidance, which is unavailable at inference time. We would like to address this issue through the following points:
>
> - Firstly, the reviewer's keen observation is absolutely correct. During inference, we **indeed mask the ground-truth texts**, resulting in the absence of textual feature guidance for our model's generation. In our practical implementation, during training, we provide the model with textual features in fusion with image encoding (as described in our method in Section 4.2). However, during inference, these textual segments are replaced by zeros, rendering them devoid of any meaningful textual "guidance" for model generation.
>
> - This naturally leads us to the question of why this textual guidance still significantly benefits our DiffuVST's performance, as empirically demonstrated in our quantitative evaluation results (shown in Section 5.2 and 5.3). We propose a hypothesis that the ground-truth text encodings, serving as guidance during model training, indirectly contribute by embedding certain training signals from the textual modality. This augmentation affects the distribution patterns within the input space of DiffuVST, which is essentially an image-conditioned text generator. Consequently, the model **learns specific attributes and distributions from the ground-truth text data**, and also **associates these features with the visual conditions**. This enhanced understanding of both the textual modality and the interplay between text and image thus empowers our model to perform better in generating natural language descriptions from visual input, as substantiated by our experimental findings.
>
> - However, we acknowledge that this discrepancy between training and inference may initially seem counterintuitive. It is indeed intriguing how the text feature guidance during training continues to exert a substantial impact on our model's generative capacity at inference time, even in the absence of explicit text guidance. We hope that our above hypothesis sheds light on this phenomenon, and our experimental evidence also encourages further exploration and analysis of this matter.
>
> In conclusion, we extend our sincere gratitude to the reviewer for posing such a valuable question. We highly appreciate its significance and will include necessary discussions on this matter in the revised version of our paper.
>
>
> **[Q9]:**
> Why do more diffusion steps result in worse performance as shown in Figure 4?
>
> **[A9]:**
> Thank you for raising this intriguing question. Although a larger number of denoising steps typically enhances generation quality, as is the usual case in image synthesis, we believe that an excessive amount of denoising iterations may lead to less satisfactory text generation, especially from the perspective of NLG metrics such as CIDEr. We attribute this phenomenon to two primary factors:
> 1. **Over-Filtering Noise:**
> Our denoising transformer is designed to predict ground-truth text embeddings $x_0$ from any noisy inputs $x_t$. Thus, even a single denoising step can effectively filter out a substantial amount of noise. When this filtering process is applied for too many iterations than necessary, however, it might inadvertently remove valuable linguistic details and lead to worse text quality.
> 1. **Loss of N-Gram Diversity:**
> Increasing the number of denoising steps may result in more uniform and less varied text, as the process continuously tries to reach a "less noisy" text representation. And metrics like CIDEr are known to lay greater weight on less common N-grams, which explains why the degrading of performance is the most noticable on the CIDEr metric.
>
> While our DiffuVST excels in generating high-quality text within only a few number of denoising steps, we acknowledge that the degradation in performance with more steps is not ideal. We appreciate the reviewer's insightful question, as it encourages our future work to explore ways for enhancing our system's stability and predictability during inference.
>
>
> **[Q10]:**
> Better to merge Tables 1 & 2 and add citations for the compared baselines.
>
> **[A10]:**
> Thank you very much for your advice. Combining Table 1 and Table 2 is indeed a good way to save space while preserving an organized representation of the results. And including citations for the baselines shown in tables will certainly provide readers with clearer references. In the revised version, we will ensure to merge Tables 1 and 2 and add the proper citations for the compared baselines. Thank you again for these constructive suggestions.
>
>
> **[Q11]:**
> Figure 2 is not very clear in terms of helping understand how the framework works.
>
> **[A11]:**
> Thank you for your valuable feedback regarding the clarity of Figure 2. We appreciate your input and would like to address your concerns with the following points:
>
> - **Clarification of Figure 2 for depicting the DiffuVST workflow:**
>   1. *Data Input and Feature Extraction:* Figure 2 begins with an image stream (depicted as five consecutive pictures at the base of Figure 2) and its corresponding story text sequences (represented by five captions within the blue box at the bottom left) as input. Our DiffuVST model then extracts both textual and visual features using BLIP text and image encoders, as well as adapter modules. This process yields in-domain textual and visual features, denoted by the orange and cyan ladder-shaped icons.
>
>   2. *Forward Diffusion Process:* The forward diffusion process involves iteratively corrupting the ground-truth caption embedding $x_0$ with Gaussian noise over $T$ diffusion stages. This transformation is depicted in the upper left section of Figure 2, where the blue triangles representing $x_0$ tokens progressively take on a noisy texture, ultimately resulting in $x_T$ after $T$ steps.
>
>   3. *Denoising Phase and Output Generation:* In the denoising process, the corrupted embedding $x_T$, along with the fused textual-visual guidance features (represented by the central light-green rectangle block), is passed to a Transformer. Specifically, we employ a BERT-like transformer-encoder architecture, as detailed in Line 324-325 of our paper. This Transformer learns to restore $x_0$, which is then rounded to produce the output text sequences, displayed in the top-right blue rectangular block.
>
> - **Our plans to improve Figure 2:** We greatly appreciate your feedback, and we are committed to making improvements to enhance the clarity of Figure 2 in the following ways:
>   - We will **add clearer annotations** to the specific elements within Figure 2, such as ladder shapes denoting adapted features, blue triangles representing ground-truth text embeddings, shaded background texture indicating noise corruption, and more.
>   - we will **supplement Figure 2 with additional figures** illustrating the iterative diffusion process and denoising phase, in order to provide a comprehensive understanding of the underlying diffusion-based mechanisms.
>   - We will also **include more detailed caption explanations** to accompany the figures, ensuring that the entire workflow of our framework is clearly conveyed.
>
>
>
> **[References]:**
> [1] Transitional Adaptation of Pretrained Models for Visual Storytelling. CVPR 2021.
> [2] BLIP: Bootstrapping Language-Image Pre-training for Unified Vision-Language Understanding and Generation. ICML 2022.
> [3] GLAC Net: GLocal Attention Cascading Networks for Multi-image Cued Story Generation. NAACL 2018.
> [4] No Metrics Are Perfect: Adversarial Reward Learning for Visual Storytelling. ACL 2018.
> [5] Hide-and-Tell: Learning to Bridge Photo Streams for Visual Storytelling. AAAI 2020.
> [6] Hierarchically Structured Reinforcement Learning for Topically Coherent Visual Story Generation. AAAI 2019.
> [7] Denoising Diffusion Probabilistic Models.2022.
> [8] Improved Denoising Diffusion Probabilistic Models. 2021.
> [9] Diffusion-LM Improves Controllable Text Generation. 2022.
> [10] WaveGrad: Estimating Gradients for Waveform Generation. 2020.

---

### Official Review · Reviewer_AmK5 · 2023-08-02

**Typos Grammar Style And Presentation Improvements:** The texts in Figures 3 and 4 are too …
**Soundness:** 3

**Excitement:**

3: Ambivalent: It has merits (e.g., it reports state-of-the-art results, the idea is nice), but there are key weaknesses (e.g., it describes incremental work), and it can significantly benefit from another round of revision. However, I won't object to accepting it if my co-reviewers champion it.

**Paper Topic And Main Contributions:**

The paper presents DiffuVST, a novel diffusion-based model for visual storytelling that aims to generate meaningful and coherent narratives from sets of pictures. The authors highlight the challenges of efficiency faced by existing autoregressive methods. To address this challenge, DiffuVST uses stochastic and non-autoregressive decoding, enabling faster and more diverse narrative generation. The model also includes history guidance and multimodal adapter modules to improve inter-sentence coherence. Experimental results on multiple fictional visual-story datasets demonstrate the superiority of DiffuVST over previous autoregressive models in terms of text quality and inference speed.

**Reasons To Accept:**

1. The proposed DiffuVST method is new, and the use of diffusion models is well motivated by inference speed. The history guidance and multimodal adapter modules are also well-motivated.
2. The authors conducted comprehensive experiments on four fictional visual-story datasets, covering story generation and story continuation tasks. The evaluation demonstrates the superior performance of DiffuVST compared to previous autoregressive models.
3. Comprehensive analyses, e.g., guidance scale, and number of inference steps, have been conducted.

**Reasons To Reject:**

The only baseline being evaluated was published in 2018. More recent methods can be compared with the proposed method.

**Reproducibility:**

4: Could mostly reproduce the results, but there may be some variation because of sample variance or minor variations in their interpretation of the protocol or method.

**Reviewer Confidence:**

3: Pretty sure, but there's a chance I missed something. Although I have a good feel for this area in general, I did not carefully check the paper's details, e.g., the math, experimental design, or novelty.

---

> ### Author Rebuttal · Authors · 2023-08-29
>
> # Responses to Reviewer AmK5
>
> Thank you for such an insightful and constructive review. We truly appreciate your comments and address your concerns and questions as follows:
>
> **[Q1]:**
> More recent baselines should be compared with the proposed method.
>
> **[A1]:**
> We appreciate the reviewer's constructive feedback regarding the need for comparing our DiffuVST with more baselines to validate its effectiveness. We acknowledge the importance of such comparisons in the context of more recent advancements in the field. To address this concern, we have incorporated additional state-of-the-art open-source auto-regressive baselines in order to strengthen the validation of our approach:
>
> 1. **TAPM [1, CVPR2021]:**
> TAPM aligns the embedding space of a pretrained visual encoder and language generator using and Adaptation Loss. Subsequently, it applies a Sequential Coherence Loss for finetuning, achieving superior VIST results over multiple visual storytelling baselines including AREL[4], INet[5], HSRL[6], etc. For our task, we utilize a pretrained ViT as the visual encoder and GPT2 for the language generator, ensuring a fair comparison, as our DiffuVST also employs a ViT encoder (BLIP-Image-Model) for image features.
>
> 2. **BLIP-EncDec [2, ICML2022]:**
> This baseline leverages both the Encoder and LM-Decoder of the pretrained BLIP captioning model, fine-tuned on the image-text pairs from our datasets. During inference, we incorporate generated captions from previous images in the textual history of the story sequence, introducing a sequential dependency for captioning the current image. Given that BLIP-EncDec builds upon a well-established captioning model, it serves as a strong baseline, with the mentioned modification specifically for visual storytelling.
>
> Considering the **accessibility** and **reproducibility** of visual storytelling models, we believe these two strong baselines, along with GLACNet[3], which was part of our initial submission, provide a comprehensive quantitative comparison for our model.
>
> The results of this extended evaluation are presented in the table below, which supplements Table 1 and 2 from our initial submission. We use "Writer Mode" as an example task setting and "DiffuVST" refers to the optimally guided version of our model, denoted as "DiffuVST_*guided*" in Table 1 and 2.
>
> | Dataset | Method | B@1 | B@4 | M | R | C | Inference_Time (sec.) ↓ |
> |   :----:  | :----:  | :----:  | :----:  | :----:  | :----:  | :----: | :----: |
> | *F.* | GLACNet [3] | 44.55 | 11.98 | 37.48 | 29.23 | 8.93 | 1.2042 |
> | *F.* | BLIP-EncDec [2] | 48.75 | 16.32 | 42.25 | 35.3 | 15.94 | 2.0486 |
> | *F.* | TAPM [1] | 49.05 | 15.08 | 42.12 | 33.03 | 16.54 | 1.8923 |
> | *F.* | DiffuVST (ours) | **49.76** | **17.01** | **42.98** | **36.22** | **17.63** | **0.0981** |
> |     |   |   |   |   |   |  |  |
> | *P.* | GLACNet [3] | 23.66 | 2.1 | 21.19 | 17.67 | 2.8 | 0.7844 |
> | *P.* | BLIP-EncDec [2] | 31.38 | 2.96 | 25.13 | 21.64 | 5.63 | 1.8702 |
> | *P.* | TAPM [1] | 28.43 | **3.78** | 24.43 | 21.87 | 5.55 | 1.2289 |
> | *P.* | DiffuVST (ours) | **32.09** | 2.81 | **25.35** | **22.03** | **6.18** | **0.1196** |
> |     |   |   |   |   |   |  |  |
> | *D.* | GLACNet [3] | 20.28 | 1.78 | 16.36 | 15.15 | 7.15 | 0.3030 |
> | *D.* | BLIP-EncDec [2] | 20.02 | 2.14 | 16.01 | 17.74 | 14.7 | 0.7474 |
> | *D.* | TAPM [1] | 20.79 | 2.12 | 16.35 | 17.68 | **15.26** | 0.4457 |
> | *D.* | DiffuVST (ours) | **22.08** | **2.74** | **17.33** | **19.75** | 15.16 | **0.0161** |
> |     |   |   |   |   |   |  |  |
> | *A.* | GLACNet [3] | 20.37 | 1.02 | 19.46 | 13.39 | 2.14 | 0.9285 |
> | *A.* | BLIP-EncDec [2] | 19.16 | **1.81** | 21.88 | 16.71 | 2.89 | 1.3141 |
> | *A.* | TAPM [1] | 21.95 | 1.56 | 21.55 | 16.3 | 2.97 | 1.1629 |
> | *A.* | DiffuVST (ours) | **23.13** | 1.41 | **21.14** | **19.07** | **3.4** | **0.1988** |
>
> As evident from the table, DiffuVST consistently outperforms these baselines **across nearly all metrics and datasets**. While there are marginal exceptions, such as the slightly lower B@4 score on the PororoSV and ASEOP datasets and the CIDEr score on DiDeMoSV compared to TAPM, these do not diminish DiffuVST's overall superior performance. Notably, given TAPM has outperformed most state-of-the-art visual storytellers in their results on VIST (as indicated in the TAPM paper[1]), we have good reason to believe that our model can also surpass these models, further demonstrating the effectiveness of DiffuVST. Moreover, it is essential to emphasize that DiffuVST's non-autoregressive nature significantly accelerates inference, making it **more than 10x times faster** than the auto-regressive baselines. This acceleration greatly enhances the efficiency of visual storytelling. In conclusion, DiffuVST not only achieves top-tier performance but also excels in efficiency, making it an overall successful choice for the task of visual storytelling.
>
> Once again, we sincerely thank the reviewer for prompting this comprehensive evaluation, and we will incorporate these additional baselines into our revised paper, further enhancing its comprehensiveness and validation of DiffuVST's capabilities.
>
>
>
> **[Q2]:**
> The texts in Figures 3 and 4 are too small to read.
>
> **[A2]:**
> Thank you for bringing this readability issue to our attention. We acknowledge that the text indeed appears too small in Figures 3 and 4. To address this, we will enlarge the text font size in these figures to ensure that all textual content is clearly visible. In addition, we also plan to provide higher-resolution versions of all our figures in the revised paper and further enhance the readability of the figures.
>
>
>
> **[References]:**
> [1] Transitional Adaptation of Pretrained Models for Visual Storytelling. CVPR 2021.
> [2] BLIP: Bootstrapping Language-Image Pre-training for Unified Vision-Language Understanding and Generation. ICML 2022.
> [3] GLAC Net: GLocal Attention Cascading Networks for Multi-image Cued Story Generation. NAACL 2018.
> [4] No Metrics Are Perfect: Adversarial Reward Learning for Visual Storytelling. ACL 2018.
> [5] Hide-and-Tell: Learning to Bridge Photo Streams for Visual Storytelling. AAAI 2020.
> [6] Hierarchically Structured Reinforcement Learning for Topically Coherent Visual Story Generation. AAAI 2019.

---

### Meta-Review · Area_Chair_5KQi · 2023-09-18

**Recommendation:** 4

**Metareview:**

**Summary evaluation**: The authors propose a diffusion based visual story telling method to improve the quality and inference speed of the task. Two active reviewers have a consensus on the novelty of the proposed method and appreciate the comprehensive analyses and evaluations. Both reviewers had similar concerns related to the limited baseline, however the authors provided additional baselines to resolve the concern from the both reviewers. Also, the authors could provide comprehensive answers to the reviewers, so the most concerns were resolved.

**Reviewer Xen1's reviews**: Reviewer Xen1 have been having a different opinion, but the reviewer haven't interacted at all and haven't even acknowledged the rebuttal by the authors. Also, it was pointed out that Xen1's summary of the paper contained exact copy from the paper which was not desirable.

**Reviewers' recommendations**: After the in-depth discussions, two active reviewers made a consensus that the soundness is good or strong. Also, in terms of excitement, both reviewers had similar consensus which is good or strong. The reviewer Xen1's scores have been '2' on both criteria since the initial review, but Xen1 haven't joined the discussion about misunderstanding of the method. Therefore, I am downweighing the opinions from Xen1.

---

### Decision · Program_Chairs · 2023-10-07

**Decision:**

Accept-Findings

**Comment:**

**Summary evaluation**: The authors propose a diffusion based visual story telling method to improve the quality and inference speed of the task. Two active reviewers have a consensus on the novelty of the proposed method and appreciate the comprehensive analyses and evaluations. Both reviewers had similar concerns related to the limited baseline, however the authors provided additional baselines to resolve the concern from the both reviewers. Also, the authors could provide comprehensive answers to the reviewers, so the most concerns were resolved.

**Reviewer Xen1's reviews**: Reviewer Xen1 have been having a different opinion, but the reviewer haven't interacted at all and haven't even acknowledged the rebuttal by the authors. Also, it was pointed out that Xen1's summary of the paper contained exact copy from the paper which was not desirable.

**Reviewers' recommendations**: After the in-depth discussions, two active reviewers made a consensus that the soundness is good or strong. Also, in terms of excitement, both reviewers had similar consensus which is good or strong. The reviewer Xen1's scores have been '2' on both criteria since the initial review, but Xen1 haven't joined the discussion about misunderstanding of the method. Therefore, I am downweighing the opinions from Xen1.